# Weston-Watkins Hinge Loss and Ordered Partitions

**Yutong Wang**
University of Michigan
yutongw@umich.edu

**Clayton D. Scott**
University of Michigan
clayscot@umich.edu

## Abstract

Multiclass extensions of the support vector machine (SVM) have been formulated in a variety of ways. A recent empirical comparison of nine such formulations [1] recommends the variant proposed by Weston and Watkins (WW), despite the fact that the WW-hinge loss is not calibrated with respect to the 0-1 loss. In this work we introduce a novel discrete loss function for multiclass classification, the *ordered partition loss*, and prove that the WW-hinge loss *is* calibrated with respect to this loss. We also argue that the ordered partition loss is *minimally emblematic* among discrete losses satisfying this property. Finally, we apply our theory to justify the empirical observation made by Doğan et al. [1] that the WW-SVM can work well even under massive label noise, a challenging setting for multiclass SVMs.

## 1 Introduction

Classification is the task of assigning labels to instances, and a common approach is to minimize misclassification error corresponding to the 0-1 loss. However, the 0-1 loss is discrete and typically cannot be optimized efficiently. To address this, the 0-1 loss is often replaced by a surrogate loss during training. If the surrogate is *calibrated* with respect to the 0-1 loss, then a classifier minimizing the expected surrogate loss will also minimize the expected 0-1 loss in the infinite sample limit.

For multiclass classification, several different multiclass extensions of the support vector machine (SVM) have been proposed, including the Weston-Watkins (WW) [2], Crammer-Singer (CS) [3], and Lee-Lin-Wahba (LLW) [4] SVMs. The pertinent difference between these multiclass SVMs is the multiclass generalization of the hinge loss. Below, we refer to the hinge loss from WW-SVM as the WW hinge loss and so on. It is well-known that the LLW-hinge is calibrated with respect to the 0-1 loss, while the WW- and CS-hinge losses are not [5, 6].

Despite this result, the LLW-SVM is not more widely accepted than the WW-, CS-, and other SVMs. The first reason for this is that while the LLW-SVM is calibrated with respect to the 0-1 loss, this did not lead to superior performance empirically. In particular, Doğan et al. [1] found that the LLW-SVM fails in low dimensional feature space even under the noiseless setting. On the other hand, Doğan et al. [1] observed that the WW-SVM is the only multiclass SVM that succeeded in both the noiseless and noisy setting in their simulations. Indeed, Doğan et al. [1] concluded that, among 9 different competing multiclass SVMs, the WW-SVM offers the best overall performance when considering accuracy and computation. The second reason is that the calibration framework is not limited to the 0-1 loss. There could be other discrete losses with respect to which a surrogate is calibrated, and which help to explain its performance. Indeed, Ramaswamy et al. [7] recently showed that the CS-hinge loss is calibrated with respect to a discrete loss for classification with abstention.

In a vein similar to [7], we show that the WW-hinge loss is calibrated with respect to a novel discrete loss that we call the *ordered partition* loss. Our results leverage the embedding framework for analyzing discrete losses and convex piecewise linear surrogates, introduced recently by Finocchiaro et al. [8]. We also give theoretical justification for the empirical performance of the WW-SVM observed by Doğan et al. [1].

## 1.1 Related work

Cortes and Vapnik [9] introduced the support vector machine for learning a binary classifier, using the hinge loss as a surrogate for the 0-1 loss. Steinwart [10] showed that the binary SVM is *universally consistent*, a desirable property of a classification algorithm that ensures its convergence to the Bayes optimal classifier in the large sample limit. Steinwart [11] later used calibration to give a more general proof of SVM consistency with respect to the 0-1 loss. Around that time, more general theories of when a loss is calibrated with respect to 0-1 loss, or "classification calibrated," began to emerge [12, 13, 14], and since then a proliferation of papers have extended these ideas to a variety of learning settings (see Bao et al. [15] for a recent review).

Several natural extensions of the binary SVM exist, including the Weston-Watkins (WW) [2], Crammer-Singer (CS) [3], and Lee-Lin-Wahba (LLW) [4] SVMs. Tewari and Bartlett [6] extended the definition of calibration with respect to the 0-1 loss to the multiclass setting. Liu [5] and Tewari and Bartlett [6] analyzed these hinge losses and showed that WW and CS hinge losses are not calibrated with respect to the 0-1 loss while the LLW hinge loss is. Doğan et al. [1] introduced a framework that unified existing multiclass SVMs, proved the 0-1 loss consistency of several multiclass SVMs when the kernel is allowed to change, and also conducted extensive experiments. Despite not being calibrated with respect to the 0-1 loss, Zhang [12] showed that the Crammer-Singer SVM is consistent given the "majority assumption", i.e., the most probable class has greater than $1/2$ probability. When the majority assumption is violated, experiments conducted by Doğan et al. [1] suggested that the CS-SVM fails, while the WW-SVM continues to perform well.

The LLW-hinge loss is calibrated with respect to the 0-1 loss while the WW-hinge loss is not [5]. Nevertheless, the WW-SVM often outperforms the LLW-SVM in experiments [1] which ostensibly undermines using calibration be as a justification for performance. To reconcile this, we refer the reader to the discussion in Doğan et al. [1, Section 3.3] on *relative* and *absolute margin* losses. Doğan et al. [1] argued that the poorer performance of losses based on absolute margin, including the LLW-hinge, is due to the issue of the absolute margin being incompatible the decision function. On the other hand, the CS and WW-hinge losses are relative margin based and do not suffer the same issue. We remark that Fathony et al. [16] proposed a relative margin hinge loss which is calibrated with respect to the 0-1 loss that outperforms the WW-hinge loss at the expense of greater computational complexity.

Ramaswamy and Agarwal [17] extended the notion of calibration to an arbitrary discrete loss used in *general multiclass learning*. The general multiclass learning framework unifies several learning problems, including cost-sensitive classification [18], classification with abstain option [7], ranking [19], and partial label learning [20]. Furthermore, Ramaswamy and Agarwal [17] introduced the concept of *convex calibration dimension* which is defined for a discrete loss to be the minimum dimension required for the domain of a convex surrogate loss to be calibrated with respect to the given discrete loss. Ramaswamy et al. [7] proved the consistency of CS-SVM with respect to the abstention loss where the cost of abstaining is $1/2$ by showing that the CS hinge is calibrated with respect to this abstention loss. They also proposed a new calibrated convex surrogate loss in dimension $\lceil \log_2 k \rceil$ for the abstention loss, implying that the CS hinge is suboptimal from the CC-dimension perspective.

Recently, several new multiclass hinge-like losses have been proposed, as well as frameworks for constructing convex losses. Doğan et al. [1] used their framework to devise two new multiclass hinge losses, and using ideas from adversarial multiclass classification, Fathony et al. [16] proposed a new multiclass hinge-like loss; all three are calibrated with respect to the 0-1 loss. Blondel et al. [21] introduced a class of losses known as *Fenchel-Young losses* which contains non-smooth losses such as the CS hinge loss as well as smooth losses such as the logistic loss. Tan and Zhang [22] proposed an approach for constructing hinge-like losses using generalized entropies. Finocchiaro et al. [8] studied the calibration properties of *polyhedral* losses using the *embedding* framework that they developed. They analyzed several polyhedral losses in the literature including the CS hinge, the Lovász hinge [23], and the top-$n$ loss [24].

## 1.2 Our contributions

We introduce a novel discrete loss $\ell$, the *ordered partition loss*. We show in Theorem 3.1 that the Weston-Watkins hinge loss $L$ embeds the ordered partition loss $\ell$. Our embedding result together with results of [8] imply that $L$ is calibrated with respect to $\ell$ (Corollary 3.2). To the best of our

knowledge, this is the first calibration-theoretic result for the WW-hinge loss. We also introduce the notion of the *minimally emblematic* discrete loss that a polyhedral loss can embed and argue that the ordered partition loss is minimally emblematic for the WW-hinge loss. In Section 5, we use properties of the ordered partition loss to give theoretical support for the empirical observations made by Doğan et al. [1] on the success of WW-SVM in the massive label noise setting.

## 1.3 Notations

Let $k \geq 3$ be an integer which denotes the number of classes. For a positive integer $n$, we let $[n] = \{1, \ldots, n\}$. If $v = (v_1, \ldots, v_k) \in \mathbb{R}^k$ and $i \in [k]$ is an index, then let $[v]_i := v_i$. Define $\max v = \max_{i \in [k]} v_i$ and $\arg \max v = \{i \in [k] : v_i = \max v\}$.

Let $\mathfrak{S}_k$ denote the set of permutations on $[k]$, i.e., elements of $\mathfrak{S}_k$ are bijections $\sigma : [k] \to [k]$. Given $\sigma \in \mathfrak{S}_k$ and $v \in \mathbb{R}^k$, the vector $\sigma v \in \mathbb{R}^k$ is defined entrywise where the $i$-th entry is $[\sigma v]_i = v_{\sigma(i)}$. Equivalently, we view $\mathfrak{S}_k$ as the set of permutation matrices in $\mathbb{R}^{k \times k}$.

Let $\mathbb{R}_+$ denote the set of nonnegative reals. Denote $\Delta^k = \{(p_1, \ldots, p_k) \in \mathbb{R}_+^k : p_1 + \cdots + p_k = 1\}$ the probability simplex. For $p \in \Delta^k$, we write $Y \sim p$ to denote a discrete random variable $Y \in [k]$ whose probability mass function is $p$. Let $\langle \cdot, \cdot \rangle$ be the usual dot-product between vectors. Denote by $\mathbb{I}\{\texttt{input}\}$ the indicator function which returns 1 if $\texttt{input}$ is true and 0 otherwise.

## 1.4 Background

Recall the *general multiclass learning* framework as described in [17]: $\mathcal{X}$ is a sample space and $P$ is a joint distribution over $\mathcal{X} \times [k]$. A *multiclass classification loss* is a function $\ell : \mathcal{R} \to \mathbb{R}_+^k$ where $\mathcal{R}$ is called the *prediction space* and $[\ell(r)]_y \in \mathbb{R}_+$ is the penalty incurred for predicting $r \in \mathcal{R}$ when the label is $y \in [k]$. If $\mathcal{R}$ is finite, we refer to $\ell$ as a *discrete loss*. For example, a common setting for classification is $\mathcal{R} = [k]$ and $\ell$ is the 0-1 loss. The $\ell$-risk of a *hypothesis* function $f : \mathcal{X} \to \mathcal{R}$ is

$$\mathrm{er}_P^\ell(f) := \mathbb{E}_{X,Y \sim P} \{[\ell(f(X))]_Y\}. \tag{1}$$

The goal is to design $\ell$-*consistent* algorithms, i.e., procedures that output a hypothesis $f_n$ based on an input of $n$ training samples sampled i.i.d from $P$ such that $\mathrm{er}_P^\ell(f_n) \to \mathrm{er}_P^{\ell,*} = \inf_{f:\mathcal{X} \to \mathcal{R}} \mathrm{er}_P^\ell(f)$ as $n \to \infty$. Since $\ell$ is discrete, eq. (1) is difficult to directly minimize. To circumvent this difficulty, we consider a convex *surrogate loss* $L : \mathbb{R}^d \to \mathbb{R}^k$ for some positive integer $d$. The following property relates the surrogate loss $L$ and the discrete loss $\ell$.

**Definition 1.1** (Calibration). For each $p \in \Delta^k$, define $\gamma_\ell(p) := \arg \min_{r \in \mathcal{R}} \langle p, \ell(r) \rangle$. We say that $L$ *is calibrated with respect to* $\ell$ if there exists a function $\psi : \mathbb{R}^d \to \mathcal{R}$ such that for all $p \in \Delta^k$

$$\inf_{u \in \mathbb{R}^d : \psi(u) \notin \gamma_\ell(p)} \langle p, L(u) \rangle > \inf_{v \in \mathbb{R}^d} \langle p, L(v) \rangle.$$

By Ramaswamy and Agarwal [17, Theorem 3], $L$ being calibrated with respect to $\ell$ is equivalent to the following: there exists $\psi : \mathbb{R}^d \to \mathcal{R}$ such that for all joint distributions $P$ on $\mathcal{X} \times [k]$ and all sequences of functions $g_n : \mathcal{X} \to \mathbb{R}^d$, we have

$$\mathrm{er}_P^L(g_n) \to \mathrm{er}_P^{L,*} \quad \text{implies} \quad \mathrm{er}_P^\ell(\psi \circ g_n) \to \mathrm{er}_P^{\ell,*}$$

where $\mathrm{er}_P^{L,*} = \inf_{g:\mathcal{X} \to \mathbb{R}^d} \mathrm{er}_P^L(g)$. Thus, the calibration property allows us to focus on finding $L$-consistent algorithms. In general it can be difficult to check that a given $L$ is calibrated with respect to $\ell$. Finocchiaro et al. [8] introduced the following definition:

**Definition 1.2** (Finocchiaro et al. [8]). The loss $L : \mathbb{R}^d \to \mathbb{R}^k$ *embeds* $\ell : \mathcal{R} \to \mathbb{R}^k$ if there exists an injection $\varphi : \mathcal{R} \to \mathbb{R}^d$ called an *embedding* such that

1. $L(\varphi(r)) = \ell(r)$ for all $r \in \mathcal{R}$

2. $r \in \arg \min_{r \in \mathcal{R}} \langle p, \ell(r) \rangle$ if and only if $\varphi(r) \in \arg \min_{v \in \mathbb{R}^d} \langle p, L(v) \rangle$.

The notion of embedding is important due to the following result from [8, Theorem 3]:

**Theorem 1.3** (Finocchiaro et al. [8]). *Let $L$ be convex piecewise-linear and $\ell$ be discrete. If $L$ embeds $\ell$, then $L$ is calibrated with respect to $\ell$.*

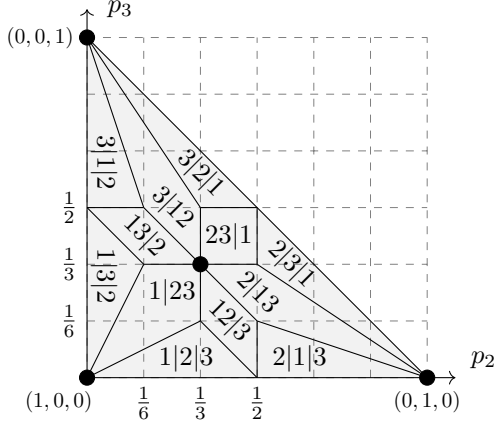

Figure 1: The gray triangle represents the probability simplex $\Delta^3$, where $(p_1, p_2, p_3) \in \Delta^3$ is plotted as $(p_2, p_3)$ in the plane. The interior of each polygonal region contains $p \in \Delta^3$ such that $\min_{\mathbf{S} \in \mathcal{OP}_k} \langle p, \ell(\mathbf{S}) \rangle$ has a unique minimizer. For the derivations, see supplementary material. Ordered partitions are represented as follows:

$$(\{1\}, \{2, 3\}) \mapsto 1|23,$$
$$(\{1\}, \{2\}, \{3\}) \mapsto 1|2|3,$$
$$\vdots$$
$$(\{3\}, \{2\}, \{1\}) \mapsto 3|2|1.$$

Given $L, \ell$ and $\varphi$, Finocchiaro et al. [8, Definition 6] provided an explicit construction for $\psi$ with excess risk bound proved in [8, Theorem 6].

In this work, we are interested in the case when $L$ is the WW-hinge loss:

**Definition 1.4.** For $v \in \mathbb{R}^k$, define the *Weston-Watkins hinge loss* [2] $L(v) \in \mathbb{R}_+^k$ entrywise by

$$[L(v)]_y = \sum_{i \in [k] \,:\, i \neq y} h(v_y - v_i), \qquad y \in [k]$$

where $h : \mathbb{R} \to \mathbb{R}_+$ is the *hinge function* defined by $h(x) = \max\{0, 1 - x\}$.

By Theorem 1.3, to prove that $L$ is calibrated with respect to $\ell$, it suffices to show that $L$ embeds $\ell$. Going forward, $L$ will refer to the WW-hinge loss. We now work toward showing that $L$ embeds the ordered partition loss $\ell$, which we introduce next.

## 2 The ordered partition loss

The prediction space $\mathcal{R}$ that we use is the set of ordered partitions, which we now define:

**Definition 2.1.** An *ordered partition* on $[k]$ of length $l$ is an ordered list $\mathbf{S} = (S_1, \ldots, S_l)$ of nonempty, pairwise disjoint subsets of $[k]$ such that $S_1 \cup \cdots \cup S_l = [k]$. Denote by $\mathcal{OP}_k$ the set of all ordered partitions on $[k]$ with length $\geq 2$. We write the length of $\mathbf{S}$ as $l_{\mathbf{S}}$ to be precise when working with multiple ordered partitions.

Ordered partitions can be thought of as a complete ranking of $k$ items where ties are allowed. They are widely studied in combinatorics [25, 26, 27]. In the ranking literature, ordered partitions are called *bucket orders* [28] and the $S_i$s are called the *buckets*. The first bucket $S_1$ contains the highest ranked items, and so on. There is only one ordered partition with $l_{\mathbf{S}} = 1$, namely the *trivial partition* $\mathbf{S} = ([k])$. Thus, $\mathcal{OP}_k$ is the set of nontrivial ordered partitions.

We now define the following discrete loss over the ordered partitions:

**Definition 2.2.** The *ordered partition loss* $\ell : \mathcal{OP}_k \to \mathbb{R}_+^k$ is defined, for $i \in [k]$ and $\mathbf{S} = (S_1, \ldots, S_l) \in \mathcal{OP}_k$, as $[\ell(\mathbf{S})]_i = |S_1| - 1 + \sum_{j=1}^{l_{\mathbf{S}}-1} |S_1 \cup \cdots \cup S_{j+1}| \cdot \mathbb{I}\{i \notin S_1 \cup \cdots \cup S_j\}$.

The intuition behind the ordered partition loss is that we want to rank the labels, where ties are allowed and each $S_i$ is a set of labels that are tied. We want the correct label to be as high up the ranking as possible. The lower the true class is ranked, the larger the loss.

To build intuitions about $\ell$, let $Y \sim p$ and consider the random variable $[\ell(\mathbf{S})]_Y$ whose expectation is

$$\mathbb{E}_{Y \sim p} \{[\ell(\mathbf{S})]_Y\} = |S_1| - 1 + \sum_{j=1}^{l_{\mathbf{S}}-1} |S_1 \cup \cdots \cup S_{j+1}| \cdot \Pr_{Y \sim p} \{Y \notin S_1 \cup \cdots \cup S_j\}. \tag{2}$$

Note that $\mathbb{E}_{Y \sim p} \{[\ell(\mathbf{S})]_Y\} = \langle p, \ell(\mathbf{S}) \rangle$. In Figure 1, we visualize the decision rule for the Bayes optimal classifier in the $k = 3$ case by plotting the function $p \mapsto \arg\min_{\mathbf{S} \in \mathcal{OP}_k} \langle p, \ell(\mathbf{S}) \rangle$. When $l_{\mathbf{S}} = 2$, we have

$$\mathbb{E}_{Y \sim p} \{[\ell(\mathbf{S})]_Y\} = |S_1| - 1 + k \Pr_{Y \sim p} \{Y \notin S_1\}. \tag{3}$$

Thus, we have a trade-off where adding elements to $S_1$ increases the $|S_1| - 1$ term but decreases the $k \Pr_{Y \sim p} \{Y \notin S_1\}$ term. More generally, when $l_{\mathbf{S}} \geq 2$, the ordered partition loss requires the predictor to associate each test instance $x$ with a nested sequence of sets $S_1, S_1 \cup S_2, \cdots$ where these sets are designed to balance the probability of containing $x$'s label with the size of the set. In the learning with partial labels settings [29, 20], for each training instance the learner observes a set of labels, one of which is the true label. The sets $S_1, S_1 \cup S_2, \ldots$ might be called *progressive partial labels* in the spirit of partial label learning [29, 20].

Next, we define the embedding that satisfies Definition 1.2 when $L$ is the WW-hinge loss and $\ell$ is the ordered partition loss:

**Definition 2.3.** The *embedding* $\varphi : \mathcal{OP}_k \to \mathbb{R}^k$ is defined as follows: Let $\mathbf{S} = (S_1, \ldots, S_l) \in \mathcal{OP}_k$. Define $\varphi(\mathbf{S}) \in \mathbb{R}^k$ entrywise so that for all $i \in [l_{\mathbf{S}}]$ and all $j \in S_i$, we have $[\varphi(\mathbf{S})]_j = -(i - 1)$.

With the discrete loss $\ell$ and the embedding map $\varphi$ defined, we now proceed to the main results.

# 3 Main results

In this work, we establish that the WW-hinge loss embeds the ordered partition loss:

**Theorem 3.1.** *The Weston-Watkins hinge loss $L : \mathbb{R}^k \to \mathbb{R}^k$ embeds the ordered partition loss $\ell : \mathcal{OP}_k \to \mathbb{R}^k$ with embedding $\varphi$ as in Definition 2.3.*

In light of Theorem 1.3, Theorem 3.1 implies

**Corollary 3.2.** *$L$ is calibrated with respect to $\ell$.*

In the remainder of this section, we develop the tools necessary to prove Theorem 3.1.

## 3.1 Vectorial representation of ordered partitions

First, we define the set $\mathfrak{S}_k \mathcal{C}_{\mathbb{Z}}$ whose elements serve as realizations of ordered partitions inside $\mathbb{R}^k$.

**Definition 3.3.** Define the following sets:

$$\mathcal{C} := \{v \in \mathbb{R}^k : v_1 = 0, v_k \leq -1, v_i - v_{i+1} \in [0, 1], \forall i \in [k - 1]\}, \qquad \mathcal{C}_{\mathbb{Z}} := \mathcal{C} \cap \mathbb{Z}^k \tag{4}$$

and finally $\mathfrak{S}_k \mathcal{C}_{\mathbb{Z}} := \bigcup_{\sigma \in \mathfrak{S}_k} \sigma \mathcal{C}_{\mathbb{Z}}$ where $\sigma \mathcal{C}_{\mathbb{Z}} = \{\sigma v : v \in \mathcal{C}_{\mathbb{Z}}\}$.

A vector $v \in \mathbb{R}^k$ is *monotonic non-increasing* if $v_1 \geq v_2 \geq \cdots \geq v_k$. Note that vectors in $\mathcal{C}_{\mathbb{Z}}$ are nonconstant, integer-valued monotonic non-increasing such that consecutive entries decrease at most by 1. Furthermore, by construction, $\mathfrak{S}_k \mathcal{C}_{\mathbb{Z}}$ consists of all possible permutations of elements in $\mathcal{C}_{\mathbb{Z}}$. Therefore, the entries of an element $v \in \mathfrak{S}_k \mathcal{C}_{\mathbb{Z}}$ take on every value in $0, -1, \ldots, -(l - 1)$ for some integer $l \in \{2, \ldots, k\}$. Thus, $v \in \mathfrak{S}_k \mathcal{C}_{\mathbb{Z}}$ can be thought of as vectorial representation of the ordered partition $\mathbf{S} = (S_1, \ldots, S_l)$ where $S_i = \{j : v_j = -(i - 1)\}$ for each $i \in [l]$. In Proposition 3.6 below, we make this notion precise.

**Lemma 3.4.** *The image of $\varphi$ is contained in $\mathfrak{S}_k \mathcal{C}_{\mathbb{Z}}$.*

*Proof.* Let $\mathbf{S} \in \mathcal{OP}_k$. It suffices to prove that there exists some $\sigma \in \mathfrak{S}_k$ such that $\sigma \varphi(\mathbf{S}) \in \mathcal{C}_{\mathbb{Z}}$. Note that by definition, we have the set of unique values of $\varphi(\mathbf{S})$ is

$$\{[\varphi(\mathbf{S})]_j : j \in [k]\} = \{0, -1, -2, \ldots, -(l_{\mathbf{S}} - 1)\}.$$

Thus, let $\sigma \in \mathfrak{S}_k$ be such that $\sigma \varphi(\mathbf{S})$ is monotonic non-increasing. Then $\sigma \varphi(\mathbf{S}) \in \mathcal{C}_{\mathbb{Z}}$. $\square$

Next, we define the inverse of $\varphi$.

**Definition 3.5.** The *quasi-link map* $\tilde{\psi} : \mathfrak{S}_k\mathcal{C}_{\mathbb{Z}} \to \mathcal{OP}_k$ is defined as follows: Given $v \in \mathfrak{S}_k\mathcal{C}_{\mathbb{Z}}$, let $l = 1 - \min_{j\in[k]} v_j$. Define $S_i = \{j \in [k] : v_j = -(i-1)\}$ for each $i \in [l]$. Finally, define $\tilde{\psi}(v) = (S_1, \ldots, S_l)$.

The tilde in $\tilde{\psi}$ is to differentiate the quasi-link from $\psi$ in Definition 1.1.

**Proposition 3.6.** *The embedding map* $\varphi : \mathcal{OP}_k \to \mathfrak{S}_k\mathcal{C}_{\mathbb{Z}}$ *given in Definition 2.3 is a bijection with inverse given by the quasi-link map* $\tilde{\psi}$ *from Definition 3.5.*

*Proof.* We first show that for all $\tilde{\psi}(\varphi(\mathbf{S})) = \mathbf{S}$ for all $\mathbf{S} = (S_1, \ldots, S_l) \in \mathcal{OP}_k$. Observe that $S_i = \{j \in [k] : [\varphi(\mathbf{S})]_j = -(i-1)\}$ for all $i = 1, 2, \ldots, l$. This implies that $\tilde{\psi}(\varphi(\mathbf{S})) = \mathbf{S}$.

Next, we show that $\varphi(\tilde{\psi}(v)) = v$ for all $v \in \mathfrak{S}_k\mathcal{C}_{\mathbb{Z}}$. Let $\mathbf{S} = (S_1, \ldots, S_l) = \tilde{\psi}(v)$. Then $[\varphi(\mathbf{S})]_j = -(i-1)$ if and only if $j \in S_i$. By definition $S_i = \{j \in [k] : v_j = -(i-1)\}$. Hence, $[\varphi(\mathbf{S})]_j = -(i-1)$ if and only if $v_j = -(i-1)$ which implies that $\varphi(\mathbf{S}) = v$, as desired. $\qquad\square$

In the next section, using $\varphi$, we prove a relationship between the inner risk functions of $L$ and $\ell$.

## 3.2 Inner risk functions

Define the *inner risk* functions $\underline{L} : \Delta^k \to \mathbb{R}_+$ and $\underline{\ell} : \Delta^k \to \mathbb{R}_+$ as follows:

$$\underline{L}(p) = \inf_{v\in\mathbb{R}^k} \langle p, L(v)\rangle, \quad \text{and} \quad \underline{\ell}(p) = \inf_{\mathbf{S}\in\mathcal{OP}_k} \langle p, \ell(\mathbf{S})\rangle. \tag{5}$$

Note that these functions appear in the second part of Definition 1.2, although here we have inf instead of min. Since $\mathcal{OP}_k$ is finite, the infimum in the definition of $\underline{\ell}$ is attained. Later, we will argue that the infimum in the definition of $\underline{L}$ is also attained.

We now state the main structural result regarding $\underline{L}$:

**Theorem 3.7.** *For all $p \in \Delta^k$, we have*

$$\underline{L}(p) = \min_{v\in\mathfrak{S}_k\mathcal{C}_{\mathbb{Z}}} \langle p, L(v)\rangle.$$

*Sketch of proof.* Note that $L$ is invariant under translation by any scalar multiple of the all ones vector. Thus, $L$ has an extra degree of freedom. We introduce a loss function $\mathcal{L} : \mathbb{R}^{k-1} \to \mathbb{R}^k$ called the *reduced WW-hinge loss*, which removes this extra degree freedom. Furthermore, there exists a mapping $\pi : \mathbb{R}^k \to \mathbb{R}^{k-1}$ such that $\langle p, L(v)\rangle = \langle p, \mathcal{L}(\pi(v))\rangle$ for all $p \in \Delta^k$ and $v \in \mathbb{R}^k$. Letting $z = \pi(v) \in \mathbb{R}^{k-1}$, we show that for a fixed $p$, the function $F_p(z) := \langle p, \mathcal{L}(z)\rangle$ is convex and piecewise-linear and the minimization of which can be formulated as a linear program [30]. Furthermore, since $F_p$ is nonnegative, the infimum $\inf_{z\in\mathbb{R}^{k-1}} F_p(z)$ is attained [30, Corollary 3.2], which implies that the infimum in the definition of $\underline{L}$ in eq. (5) is attained as well. The linear program is shown to be totally unimodular, which implies that an integral solution exists [31], i.e., $\min_{z\in\mathbb{R}^{k-1}} F_p(z) = F_p(z^*)$ for some $z^* \in \mathbb{Z}^{k-1}$. From $z^*$, we obtain an integral $v^* \in \mathbb{Z}^k$ such that $\underline{L}(p) = \langle p, L(v^*)\rangle$. Finally, we construct an element $v^\dagger \in \mathfrak{S}_k\mathcal{C}_{\mathbb{Z}}$ from $v^*$ in such a way that the objective does not increase, i.e., $\langle p, L(v^*)\rangle \geq \langle p, L(v^\dagger)\rangle$, which implies that $\underline{L}(p) = \langle p, L(v^\dagger)\rangle$ by the optimality of $v^*$. $\qquad\square$

The ordered partition loss $\ell$ and the WW-hinge loss $L$ are related by the following:

**Theorem 3.8.** *For all $p \in \Delta^k$ and all $\mathbf{S} \in \mathcal{OP}_k$, we have*

$$\langle p, \ell(\mathbf{S})\rangle = \langle p, L(\varphi(\mathbf{S}))\rangle,$$

*where $\varphi$ is the embedding map as in Definition 2.3.*

*Sketch of proof.* Let $\mathbf{S} = (S_1, \ldots, S_l) \in \mathcal{OP}_k$ and $p \in \Delta^k$. Let $T \in \mathbb{R}^{k\times k}$ consist of ones on and below the main diagonal and zero everywhere else. Letting $D = T^{-1}$, we have

$$\langle p, L(\varphi(\mathbf{S}))\rangle = \langle p, TDL(\varphi(\mathbf{S}))\rangle = \langle T'p, DL(\varphi(\mathbf{S}))\rangle.$$

Next, we observe that $[T'p]_i = p_i + \cdots + p_k$ for each $i \in [k]$. We then show through a lengthy calculation that for each $i \in [k]$

1. If $i = 1$, then $[T'p]_1 = 1$ and $[DL(\varphi(\mathbf{S}))]_1 = |S_1| - 1$.

2. If $i > 1$ and $i = |S_1 \cup \cdots \cup S_j| + 1$ for some $j \in [l]$, then $[T'p]_i = \Pr_{Y \sim p}\{Y \notin S_1 \cup \cdots \cup S_j\}$ and $[DL(\varphi(\mathbf{S}))]_i = |S_1 \cup \cdots \cup S_{j+1}|$.

3. For all other $i$, $[DL(\varphi(\mathbf{S}))]_i = 0$ (in which case the value of $[T'p]_i$ is irrelevant).

From this, we deduce that $\langle T'p, DL(\varphi(\mathbf{S}))\rangle$ is equal to eq. (2). $\qquad\square$

Next, we show that the inner risks of $L$ and $\ell$ from eq. (5) are in fact identical:

**Corollary 3.9.** *For all $p \in \Delta^k$, we have $\underline{L}(p) = \underline{\ell}(p)$.*

*Proof.* Observe that

$$\underline{\ell}(p) \stackrel{(a)}{=} \min_{\mathbf{S} \in \mathcal{OP}_k} \langle p, \ell(\mathbf{S})\rangle \stackrel{(b)}{=} \min_{\mathbf{S} \in \mathcal{OP}_k} \langle p, L(\varphi(\mathbf{S}))\rangle \stackrel{(c)}{=} \min_{v \in \mathfrak{S}_k \mathcal{C}_{\mathbb{Z}}} \langle p, L(v)\rangle \stackrel{(d)}{=} \underline{L}(p)$$

where $(a)$ follows from definition of $\underline{\ell}$, $(b)$ from Theorem 3.8, $(c)$ from the fact that $\varphi : \mathcal{OP}_k \to \mathfrak{S}_k \mathcal{C}_{\mathbb{Z}}$ is a bijection (Proposition 3.6), and $(d)$ from Theorem 3.7. $\qquad\square$

Having developed all the tools necessary, we turn toward the proof of our main result Theorem 3.1.

### 3.3 Proof of Theorem 3.1

We check that the two conditions in Definition 1.2 holds. The first condition is that $L(\varphi(\mathbf{S})) = \ell(\mathbf{S})$ for all $\mathbf{S} \in \mathcal{OP}_k$, which follows from Theorem 3.8. To see this, note that for all $i \in [k]$ the $i$-th elementary basis vector $e_i \in \Delta^k$. Thus, we have

$$[L(\varphi(\mathbf{S}))]_i = \langle e_i, L(\varphi(\mathbf{S}))\rangle = \langle e_i, \ell(\mathbf{S})\rangle = [\ell(\mathbf{S})]_i$$

for all $i \in [k]$. This implies that $L(\varphi(\mathbf{S})) = \ell(\mathbf{S})$, which is the first condition of Definition 1.2.

Next, we check the second condition. Let $p \in \Delta^k$. Define $\gamma(p) := \arg\min_{\mathbf{S} \in \mathcal{OP}_k} \langle p, \ell(\mathbf{S})\rangle$, and $\Gamma(p) := \arg\min_{v \in \mathbb{R}^k} \langle p, L(v)\rangle$. Furthermore, by the definition of $\gamma$, $\mathbf{S} \in \gamma(p)$ if and only if $\langle p, \ell(\mathbf{S})\rangle = \underline{\ell}(p)$. Likewise, $\varphi(\mathbf{S}) \in \Gamma(p)$ if and only if $\langle p, L(\varphi(\mathbf{S}))\rangle = \underline{L}(p)$. By Corollary 3.9 and Theorem 3.8, we have $\langle p, \ell(\mathbf{S})\rangle = \underline{\ell}(p)$ if and only if $\langle p, L(\varphi(\mathbf{S}))\rangle = \underline{L}(p)$. Putting it all together, we get $\mathbf{S} \in \gamma(p)$ if and only if $\varphi(\mathbf{S}) \in \Gamma(p)$, which is the second condition of Definition 1.2.

## 4 Minimially emblematic losses

Going forward, let $L : \mathbb{R}^d \to \mathbb{R}^k_+$ be a generic surrogate loss. The WW-hinge loss is denoted by $L^{WW}$ and the CS-hinge loss by $L^{CS}$. Likewise, let $\ell : \mathcal{R} \to \mathbb{R}^k_+$ be a generic discrete loss. The ordered partition loss is denoted by $\ell^{\mathcal{OP}}$ and the 0-1 loss by $\ell^{zo}$.

We define a "dual" notion to the embedding dimension Finocchiaro et al. [32, Definition 6]:

**Definition 4.1.** Let $L : \mathbb{R}^d \to \mathbb{R}^k_+$ be a loss. Define the *embedding cardinality* of $L$ as

$$\text{emb.card}(L) := \min\left\{n \in \{2, 3, \dots\} \mid \begin{array}{c} \text{there exists a discrete loss } \ell_{:[n] \to \mathbb{R}^k} \\ \text{such that } L \text{ embeds } \ell \end{array}\right\}.$$

A discrete loss $\ell : \mathcal{R} \to \mathbb{R}^k$ is said to be *minimally emblematic* for $L$ if $|\mathcal{R}| = \text{emb.card}(L)$ and $L$ embeds $\ell$.

*Remark 4.2.* Intuitively, $\ell$ is minimally emblematic for $L$ with embedding $\varphi$ if $\varphi(\mathcal{R})$ captures all the *essential information* contained in the surrogate L in the most compact way. Let us say that a set of vectors $E \subseteq \mathbb{R}^k$ is an *emblem* of $L$ if for all $p \in \Delta^k$, the set $E \cap \arg\min_v \langle p, L(v)\rangle$ is nonempty. Then we can equivalently define $\ell$ with $\varphi$ to be *minimally emblematic* for $L$ if $\varphi(\mathcal{R})$ is an emblem of $L$ of minimal cardinality. In other words, $\varphi(\mathcal{R})$ is a minimal set of minimizers of all possible $L$-inner risks.

For each $k \in \{3, \dots, 15\}$, we showed by a computer search that for all $\mathbf{S} \in \mathcal{OP}_k$, there exists $p \in \Delta^k$ such that $\mathbf{S}$ is the *unique* minimizer of $\min_{\mathbf{T} \in \mathcal{OP}_k} \langle p, \ell(\mathbf{T})\rangle$. A consequence of this is that

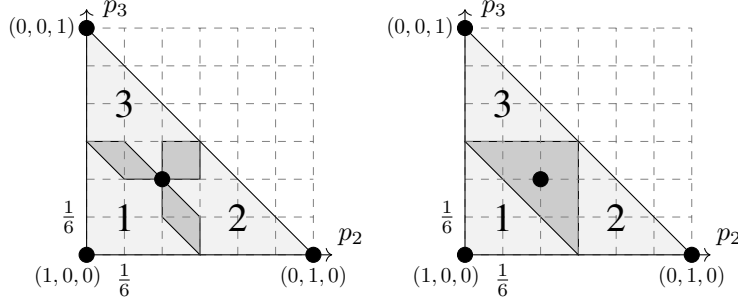

Figure 2: The gray triangle represents the probability simplex $\Delta^3$, where $(p_1, p_2, p_3) \in \Delta^3$ is plotted as $(p_2, p_3)$ in the plane. The light gray regions are $\Omega_{L^{ww}}$ (left) and $\Omega_{L^{cs}}$ (right). For the derivation, see Supplementary Material.

**Proposition 4.3.** *For $k \in \{3, \ldots, 15\}$, $\mathrm{emb.card}(L^{WW}) = |\mathcal{OP}_k|$. In other words, the ordered partition loss is minimally emblematic for the WW-hinge loss.*

We conjecture this result holds for all $k \geq 3$.

## 5 The argmax link

Define $\gamma_\ell(p) := \arg\min_{r \in \mathcal{R}} \langle p, \ell(r) \rangle$ and $\Gamma_L(p) := \arg\min_{v \in \mathbb{R}^d} \langle p, L(v) \rangle$. For multiclass classification into $k$ classes, most multiclass SVMs typically output a vector of scores $v \in \mathbb{R}^k$ which is converted to a class label by taking $\arg\max v$. In this section, we analyze the $\arg\max$ as a "link" function. Recall from Section 1.3, $\arg\max$ is a set-valued function. Define

$$\Omega_L := \{p \in \Delta^k : |\arg\max p| = 1, \arg\max v = \arg\max p, \forall v \in \Gamma_L(p)\}.$$

When $L$ is calibrated with respect to $\ell^{zo}$, we have that $\Omega_L = \{p \in \Delta^k : |\arg\max p| = 1\}$. Hence, $\Delta^k \setminus \Omega_L$ has measure zero. For other $L$ not necessarily calibrated with respect to $\ell^{zo}$, it is desirable that $\Omega_L$ be as large as possible. Below, we will prove that $\Omega_{L^{cs}}$ is a proper subset of $\Omega_{L^{ww}}$.

Recall that $\mathcal{X}$ is a sample space and $P$ is a distribution on $\mathcal{X} \times [k]$. For each $x \in \mathcal{X}$, define the *class conditional distribution* $\eta_P(x) \in \Delta^k$ by $[\eta_P(x)]_y = \Pr_{X,Y \sim P}(Y = y | X = x)$.

**Proposition 5.1.** *Let $P$ be a joint distribution on $\mathcal{X} \times [k]$ such that $\eta_P(x) \in \Omega_L$ for all $x$ and $L : \mathbb{R}^d \to \mathbb{R}_+^k$ be a loss. Let $g^* : \mathcal{X} \to \mathbb{R}^k$ be such that $g^*(x) \in \Gamma_L(\eta_P(x))$ for all $x \in \mathcal{X}$. Then $\arg\max \circ g^*$ is Bayes optimal with respect to the 0-1 loss.*

*Proof.* By definition of $\Omega_L$, we have $\arg\max \circ g^*(x) = \arg\max \eta_P(x)$ for all $x \in \mathcal{X}$. $\square$

The following theorem asserts that for any $v \in \Gamma_{L^{ww}}(p)$, the $\arg\max v$ is contained in the top bucket $S_1$ for some $\mathbf{S} \in \gamma_{\ell^{\mathcal{OP}}}(p)$.

**Theorem 5.2.** *Let $p \in \Delta^k$ be such that $\max p > \frac{1}{k}$ and $v \in \Gamma_{L^{ww}}(p)$. Then there exists $\mathbf{S} = (S_1, \ldots, S_l) \in \gamma_{\ell^{\mathcal{OP}}}(p)$ such that $\arg\max v \subseteq S_1$.*

Below, we consider two conditions on $p \in \Delta^k$ such that for *all* $\mathbf{S} \in \gamma_{\ell^{\mathcal{OP}}}(p)$, the top bucket $S_1 = \arg\max p$. By Theorem 5.2, for such $p \in \Delta^k$, we can recover $\arg\max p$ from any $v \in \Gamma_{L^{ww}}(p)$. The first condition covers $p \in \Delta^k$ such that the top class has a majority:

**Proposition 5.3.** *Let $p \in \Delta^k$ satisfy the "majority condition": $\max p > 1/2$. Then for all $\mathbf{S} = (S_1, \ldots, S_l) \in \gamma_{\ell^{\mathcal{OP}}}(p)$, we have $|S_1| = 1$ and $S_1 = \arg\max p$.*

While Proposition 5.3 does not guarantee that $\gamma_{\ell^{\mathcal{OP}}}(p)$ is a singleton, all $\mathbf{S} \in \gamma_{\ell^{\mathcal{OP}}}(p)$ have the same top bucket. The second condition covers $p \in \Delta^k$ whose top class may not have a majority, yet $\arg\max p$ can still be recovered from any $v \in \Gamma_{L^{ww}}(v)$ by taking $\arg\max v$:

**Proposition 5.4.** *Fix a number $\alpha$ such that $1 > \alpha > \frac{1}{k}$. Let $p \in \Delta^k$ satisfy the "symmetric label noise (SLN) condition": there exists $j^* \in [k]$ so that $p_{j^*} = \alpha$ and $p_j = \frac{1-\alpha}{k-1}$ for all $j \neq j^*$. Then $(\{j^*\}, [k] \setminus \{j^*\})$ is the unique element of $\gamma_{\ell^{OP}}(p)$.*

In particular, when $\alpha < 1/2$, $p$ violates the majority condition. Under SLN, we have $\arg\max p = \{j^*\}$ since $\alpha - \frac{1-\alpha}{k-1} = \frac{(k-1)\alpha-1+\alpha}{k-1} = \frac{k\alpha-1}{k-1} > \frac{1-1}{k-1} = 0$. In light of Theorem 5.2, we have

**Corollary 5.5.** *If $p \in \Delta^k$ satisfies the majority or the SLN condition, then $p \in \Omega_{L^{WW}}$.*

Thus, in two common regimes where for all $x \in \mathcal{X}$ the class conditional $\eta_P(x)$ satisfies the SLN or the majority condition, the Bayes optimal ordered partition has a top bucket consisting of a single element. When this occurs, the argmax link recovers the most probable class, i.e., the unique element from the top bucket. This supports the observation by Doğan et al. [1] that the WW-SVM performs well under the SLN condition, even with significant label noise. For the CS-hinge loss, it is known that $\Omega_{L^{CS}} = \{p \in \Delta^k : p \text{ satisfies the majority condition}\}$ [5, Lemma 4]. In particular, $\Omega_{L^{CS}}$ is a proper subset of $\Omega_{L^{WW}}$. For $k = 3$, we show in Figure 2 the regions $\Omega_{L^{WW}}$ and $\Omega_{L^{CS}}$. Our finding provides theoretical support for the finding of [1] that WW outperform CS.

# 6 Conclusion and future work

We proved that the Weston-Watkins hinge loss is calibrated with respect to the ordered partition loss, which we argue is minimally emblematic for the WW-hinge loss. Furthermore, we showed the advantage of WW-hinge loss over the Crammer-Singer hinge loss when the popular "argmax" link is used. An interesting direction is to apply the ordered partition loss to other multiclass learning problems such as partial label and multilabel learning.

# Broader Impact

This work does not present any foreseeable societal consequence.

# Acknowledgements

The authors were supported in part by the National Science Foundation under awards 1838179 and 2008074, by the Department of Defense, Defense Threat Reduction Agency under award HDTRA1-20-2-0002, and by the Michigan Institute for Data Science.

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
