[Supplementary Material]

# Supplementary material:
# Weston-Watkins Hinge Loss and Ordered Partitions

**Yutong Wang**
University of Michigan
yutongw@umich.edu

**Clayton D. Scott**
University of Michigan
clayscot@umich.edu

## Contents

# S1 Organization of the contents

Sections S3 to S5 contain the proofs for all results stated in the matching sections from the main article.

References to contents in the supplementary material are prefixed with an "S", e.g., Lemma S3.20, eq. (S7), Section S2, and Figure S1.

References to contents in the main article do not have any prefix, e.g., Theorem 1.3 and Section 1.4.

The main article and the supplementary material use two distinct bibliographies with different numberings for citations.

# S2 Notations

We introduce notations in addition to those already defined in in the main article's Section 1.3.

- $L$ always denotes the WW-hinge loss (Definition 1.4) and $\ell$ always denotes the ordered partition loss (Definition 2.2). The main article sometimes works with generic losses $L$ and $\ell$. However, the supplementary material focuses exclusively on the WW-hinge and the ordered partition loss. The exception is the last section Section S6.2, where the explicit names $L^{WW}$ and $L^{CS}$ are used.

- All vectors are column vectors unless stated otherwise.

- $\mathbb{R}_+$ and $\mathbb{Z}_+$ denotes the set of non-negative reals and integers, respectively.

- Define $\mathbb{R}_\uparrow^k = \{v \in \mathbb{R}^k : v_1 \leq v_2 \leq \cdots \leq v_k\}$. Likewise, define $\mathbb{R}_\downarrow^k$.

- For a positive integer $n$, we let $[n] := \{1, \ldots, n\}$. By convention, $[0] = \emptyset$.

- Let $\mathbf{1}^k \in \mathbb{R}^k$ denote the vector all ones.

- For a number $t \in \mathbb{R}$, let $[t]_+ = \max\{0, t\}$. For a vector $v$, we denote by $[v]_+$ the vector resulting from applying $[\cdot]_+$ entrywise to $v$. The *hinge loss* $h : \mathbb{R} \to \mathbb{R}_+$ is defined by $h(x) = [1 - x]_+$.

- For a vector $v \in \mathbb{R}^k$, we use $[v]_i$ to denote the $i$-th entry of $v$ in conjunction with the usual notation $v_i$.

- Given a vector $v \in \mathbb{R}^k$, we define

$$\max v := \max_{i \in [k]} v_i \quad \text{and} \quad \arg\max v := \{i \in [k] : v_i = \max v\}$$

Define $\min v$ and $\arg\min v$ likewise.

- Probability simplex

$$\Delta^k = \{p = (p_1, \ldots, p_k) \in \mathbb{R}_+^k : p_1 + \cdots + p_k = 1\}$$

and *non-increasing* probability simplex

$$\Delta_\downarrow^k = \{p \in \Delta^k : p_1 \geq p_2 \geq \cdots \geq p_k\} = \Delta^k \cap \mathbb{R}_\downarrow^k.$$

- For $p \in \Delta^k$, we write $Y \sim p$ to denote a discrete random variable $Y \in [k]$ whose probability mass function is $p$.

- For each $i, j \in [k]$, $\sigma_{(i,j)} \in \mathbb{R}^{k \times k}$ is the permutation matrix that switches the $i$-th and $j$-th index. By convention, if $i = j$, then $\sigma_{(i,j)}$ is the identity. Also, for brevity, define $\sigma_i = \sigma_{(1,i)}$.

- According to the definition above, $\sigma_{(i,j)}$ acts on $\mathbb{R}^k$. However, we abuse notation and allow $\sigma_{(i,j)}$ to act on $[k]$ in the obvious way. In such cases, we write $\sigma_{(i,j)}(\ell)$ for $\ell \in [k]$.

## S3 Main results

**Lemma S3.1.** *For all $v \in \mathbb{R}^k$ and $c \in \mathbb{R}$, we have $L(v) = L(v + c\mathbf{1}^k)$.*

*Proof.* For all $y \in [k]$, we have that

$$[L(v + c\mathbf{1})]_y = \sum_{i \in [k] : i \neq y} h(v_y + c - (v_i - c)) = \sum_{i \in [k] : i \neq y} h(v_y - v_i) = [L(v)]_y.$$

$\square$

**Lemma S3.2.** *For all $j \in [k]$, we have $L(\sigma_j v) = \sigma_j L(v)$.*

*Proof.* If $j = 1$, then the result is trivial. Hence, let $j > 1$. We prove

$$[L(\sigma_j v)]_y = [L(v)]_{\sigma_j(y)}$$

for the following three cases: $y \notin \{1, j\}$, $y = 1$ and $y = j$. Before we go through the cases, note that

$$[L(\sigma_j v)]_y = \sum_{i \in [k]: i \neq y} h([\sigma_j v]_y - [\sigma_j v]_i) = \sum_{i \in [k]: i \neq y} h(v_{\sigma_j(y)} - v_{\sigma_j(i)}).$$

Now, for the first case, suppose that $y \notin \{1, j\}$. Then $\sigma_j(y) = y$ and so

$$
\begin{aligned}
[L(\sigma_j v)]_y &= \sum_{i \in [k]: i \neq y} h(v_y - v_{\sigma_j(i)}) \\
&= h(v_y - v_{\sigma_j(1)}) + h(v_y - v_{\sigma_j(j)}) + \sum_{i \in [k]: i \notin \{1,j,y\}} h(v_y - v_{\sigma_j(i)}) \\
&= h(v_y - v_j) + h(v_y - v_1) + \sum_{i \in [k]: i \notin \{1,j,y\}} h(v_y - v_i) \\
&= \sum_{i \in [k]: i \notin \{y\}} h(v_y - v_i) \\
&= [L(v)]_y = [L(v)]_{\sigma_j(y)}.
\end{aligned}
$$

Next, suppose that $y = 1$. Thus, we have $\sigma_j(y) = \sigma_j(1) = j$. So

$$
\begin{aligned}
[L(\sigma_j v)]_y = [L(\sigma_j v)]_1 &= \sum_{i \in [k]: i \neq 1} h(v_j - v_{\sigma_j(i)}) \\
&= \sum_{i \in [k]: i \neq j} h(v_j - v_i) \\
&= [L(v)]_j = [L(v)]_{\sigma_j(y)}.
\end{aligned}
$$

Finally, if $y = j$, $\sigma_j(y) = 1$

$$
\begin{aligned}
[L(\sigma_j v)]_y = [L(\sigma_j v)]_j &= \sum_{i \in [k]: i \neq j} h(v_1 - v_{\sigma_j(i)}) \\
&= \sum_{i \in [k]: i \neq 1} h(v_j - v_i) \\
&= [L(v)]_1 = [L(v)]_{\sigma_j(j)} = [L(v)]_{\sigma_j(y)}.
\end{aligned}
$$

$\square$

**Lemma S3.3.** *Let $i, j \in \{2, \ldots, k\}$ be distinct. Then $\sigma_i \sigma_j \sigma_i = \sigma_{(i,j)}$.*

*Proof.* This is simply an exhaustive case-by-case proof over all inputs $y \in [k]$. First, let $y = 1$. Then $\sigma_{(i,j)}(1) = 1$ since $1 \notin \{i, j\}$. On the other hand $\sigma_i \sigma_j \sigma_i(1) = \sigma_i \sigma_j(i) = \sigma_i(i) = 1$. Now, let $y \in \{2, \ldots, k\}$. If $y \notin \{i, j\}$, then $\sigma_{(i,j)}(y) = y$ and $\sigma_i \sigma_j \sigma_i(y) = \sigma_i \sigma_j(y) = \sigma_i(y) = y$. If $y = i$, then $\sigma_{(i,j)}(i) = j$ and $\sigma_i \sigma_j \sigma_i(i) = \sigma_i \sigma_j(1) = \sigma_i(j) = j$. If $y = j$, then $\sigma_{(i,j)}(j) = i$ and $\sigma_i \sigma_j \sigma_i(j) = \sigma_i \sigma_j(j) = \sigma_i(1) = i$. $\square$

**Corollary S3.4.** *Every $\sigma \in \mathfrak{S}_k$ can be written as a product $\sigma = \sigma_{i_1}\sigma_{i_2}\cdots\sigma_{i_l}$.*

*Proof.* We prove the equivalent statement that the set $\mathcal{S} := \{\sigma_i : i \in \{2,\ldots,k\}\}$ generates the group $\mathfrak{S}_k$. A standard result in group theory states that the set of transpositions $\mathcal{T}$ generates $\mathfrak{S}_k$. By Lemma S3.3, transpositions between labels in $\{2,\ldots,k\}$ can be generated by $\mathcal{S}$. Furthermore, $\sigma_i = \sigma_{(1,i)}$ by definition, so transposition between 1 and elements of $\{2,\ldots,k\}$ can be generated by $\mathcal{S}$ as well. Hence, all of $\mathcal{T}$ can be generated by $\mathcal{S}$. $\square$

**Corollary S3.5.** *For all $v \in \mathbb{R}^k$ and $\sigma \in \mathfrak{S}_k$, we have*
$$L(\sigma v) = \sigma L(v).$$

*Proof.* By Corollary S3.4, we may write $\sigma = \sigma_{i_1}\sigma_{i_2}\cdots\sigma_{i_m}$. Hence,

$$L(\sigma v) = L(\sigma_{i_1}\sigma_{i_2}\cdots\sigma_{i_m}v) \tag{S1}$$
$$= \sigma_{i_1}L(\sigma_{i_2}\cdots\sigma_{i_m}v) \tag{S2}$$
$$\vdots$$
$$= \sigma_{i_1}\sigma_{i_2}\cdots\sigma_{i_m}L(v) \tag{S3}$$
$$= \sigma L(v), \tag{S4}$$

where for eq. (S2) to eq. (S3) we used Lemma S3.2. $\square$

**Lemma S3.6.** *Let $v \in \mathbb{R}^k$ and $j, j' \in [k]$ be distinct such that $v_j \geq v_{j'}$. Then $[L(v)]_j \leq [L(v)]_{j'}$. Furthermore, if $v_j > v_{j'}$, then $[L(v)]_j < [L(v)]_{j'}$.*

*Proof.* We have

$$[L(v)]_j - [L(v)]_{j'}$$
$$= \sum_{i\in[k]:i\neq j} h(v_j - v_i)$$
$$\quad - \sum_{i\in[k]:i\neq j'} h(v_{j'} - v_i)$$
$$= h(v_j - v_{j'}) + \sum_{i\in[k]:i\notin\{j,j'\}} h(v_j - v_i)$$
$$\quad - h(v_{j'} - v_j) - \sum_{i\in[k]:i\notin\{j,j'\}} h(v_{j'} - v_i)$$
$$= h(v_j - v_{j'}) - h(v_{j'} - v_j)$$
$$\quad + \sum_{i\in[k]:i\notin\{j,j'\}} h(v_j - v_i) - h(v_{j'} - v_i).$$

Since and $h$ is monotonically non-increasing, we have

$$v_j - v_{j'} \geq 0 \geq v_{j'} - v_j \implies h(v_j - v_{j'}) - h(v_{j'} - v_j) \leq 0 \tag{S5}$$

For the same reason, we have $h(v_j - v_i) - h(v_{j'} - v_i) \leq 0$. Putting it all together, we have $[L(v)]_j - [L(v)]_{j'} \leq 0$, as desired.

For the "furthermore" part, note that under the assumption $v_j > v_{j'}$, all inequalities in eq. (S5) becomes strict. $\square$

For reasons that will become clear later, we define for each $n \in [k-1]$

$$\underline{L}^n(p) := \inf_{v\in\mathbb{R}^k\,:\,|\arg\max v|\geq n} \langle p, L(v)\rangle. \tag{S6}$$

Since $\arg\max v$ is always nonempty, the condition that $|\arg\max v| \geq 1$ is always true. Thus, we have $\underline{L}^1 = \underline{L}$.

**Lemma S3.7.** *For all $n \in [k-1]$, $p \in \Delta^k$ and $\sigma \in \mathfrak{S}_k$, we have $\underline{L}^n(p) = \underline{L}^n(\sigma p)$.*

*Proof.* Define $\mathcal{R}^{k,n} := \{v \in \mathbb{R}^k : |\arg\max v| \geq n\}$. Since $|\arg\max v| = |\arg\max \sigma v|$, we have $\sigma\mathcal{R}^{k,n} = \mathcal{R}^{k,n}$. Introducing the change of variables $u = \sigma v$, we have

$$
\begin{aligned}
\underline{L}^n(p) &= \inf_{v \in \mathcal{R}^{k,n}} \langle p, L(v) \rangle \\
&= \inf_{\sigma' u \in \mathcal{R}^{k,n}} \langle p, L(\sigma' u) \rangle \quad \because \text{Definition of } u \\
&= \inf_{u \in \sigma \mathcal{R}^{k,n}} \langle p, L(\sigma' u) \rangle \quad \because \sigma^{-1} = \sigma' \\
&= \inf_{u \in \mathcal{R}^{k,n}} \langle p, L(\sigma' u) \rangle \quad \because \sigma\mathcal{R}^{k,n} = \mathcal{R}^{k,n} \\
&= \inf_{u \in \mathcal{R}^{k,n}} \langle p, \sigma' L(u) \rangle \quad \because \text{Corollary S3.5} \\
&= \inf_{u \in \mathcal{R}^{k,n}} \langle \sigma p, L(u) \rangle \\
&= \underline{L}^n(\sigma p).
\end{aligned}
$$

$\square$

**Lemma S3.8.** *Let* $p \in \mathbb{R}^k_\downarrow$, $q \in \mathbb{R}^k$ *be arbitrary and* $\sigma \in \mathfrak{S}_k$ *be such that* $\sigma q \in \mathbb{R}^k_\uparrow$. *Then* $\langle p, q \rangle \geq \langle p, \sigma q \rangle$.

*Proof.* Consider the "bubble sort" algorithm applied to $q$:

1. Initialize $q^{(0)} = q$, $t \leftarrow 0$

2. While there exists $i \in [k-1]$ such that $q_i^{(t)} > q_{i+1}^{(t)}$, do

   (a) $q^{(t+1)} \leftarrow \sigma_{(i,i+1)} q^{(t)}$
   (b) $t \leftarrow t+1$

3. Output monotone non-decreasing vector $q^{(t)}$

We claim that at every step, we have $\langle p, q^{(t)} \rangle \geq \langle p, q^{(t+1)} \rangle$. Let $a = q_i^{(t)}$ and $b = q_{i+1}^{(t)}$ as in step 2 above. Let $c = p_i$ and $d = p_{i+1}$. Hence, we have $a > b$ and $c \geq d$. Observe that

$$
\langle p, q^{(t)} \rangle - \langle p, q^{(t+1)} \rangle = ac + bd - (ad + bc) = (a-b)(c-d) \geq 0
$$

which proves the claim. Thus, we have

$$
\langle p, q \rangle = \langle p, q^{(0)} \rangle \geq \langle p, q^{(1)} \rangle \geq \cdots \geq \langle p, q^{(t)} \rangle.
$$

By construction, there exists $\tau \in \mathfrak{S}_k$ such that $\tau q = q^{(t)}$. We must have $\tau q = \sigma q$ since both vectors are monotone non-increasing, although $\tau$ may not equal $\sigma$. $\square$

Define the matrix $T \in \mathbb{R}^{k \times k}$

$$
T_{ij} = \begin{cases} 1 & i \geq j \\ 0 & \text{otherwise.} \end{cases} \tag{S7}
$$

Also, define $D \in \mathbb{R}^{k \times k}$

$$
D_{ij} = \begin{cases} 1 & : i = j \\ -1 & : i = j+1 \\ 0 & : \text{otherwise.} \end{cases}
$$

In other words, $D$ is the matrix with 1s on the main diagonal, $-1$s on the subdiagonal below the main diagonal, and 0 everywhere else. We have

$$
[Dv]_i = \begin{cases} v_1 & : i = 1 \\ v_i - v_{i-1} & : i > 1. \end{cases}
$$

**Lemma S3.9.** $D^{-1} = T$.

*Proof.* Using Gaussian elimination for inverting a matrix, it is easy to see that $D'T'$ is the identity. $\square$

**Definition S3.10.** Define the following sets:

$$\mathcal{M} = \{v \in \mathbb{R}^k : v_1 = 0 \text{ and } 0 \le v_i - v_{i+1}, \forall [k-1]\},$$

$$\mathcal{C} = \{v \in \mathbb{R}^k : v_1 = 0, v_k \le -1, \text{ and } 0 \le v_i - v_{i+1} \le 1, \forall [k-1]\},$$

$\mathcal{M}_\mathbb{Z} = \mathcal{M} \cap \mathbb{Z}^k$ and $\mathcal{C}_\mathbb{Z} = \mathcal{C} \cap \mathbb{Z}^k$.

**Lemma S3.11.** *We have the following equality of sets:*

$$\mathcal{M}_\mathbb{Z} = \{-Tc : c \in \mathbb{Z}_+^k, \ c_1 = 0\}$$
$$\mathcal{C}_\mathbb{Z} = \{-Ts : s \in \{0,1\}^k, \ s_1 = 0, \ and \ \exists i \in \{2, \dots, k\} : s_i = 1\}$$

*Proof.* If $v \in \mathcal{C}_\mathbb{Z}$, then we have $v_i \in \mathbb{Z}_+$ and $v_i - v_{i+1} \in [0,1]$. These two conditions together implies that $v_i - v_{i+1} \in \{0,1\}$ for all $i \in [k-1]$. Hence, $-Dv \in \{0,1\}^{k-1}$ with $[Dv]_1 = -v_1 = 0$. Let $-Dv = s$. Then Lemma S3.9 implies that $-Ts = TDv = v$. By construction, $s_1 = 0$. Furthermore, if $s_i = 0$ for all $i \in [k]$, then we would have $v = 0$ as well, which contradicts the fact that $v_k \le -1$. Hence, there must exists $i \in \{2, \dots, k\}$ such that $s_i = 1$. Clearly, all $v \in \mathcal{C}_\mathbb{Z}$ arise this way. The statement about $\mathcal{M}_\mathbb{Z}$ is similar. □

**Lemma S3.12.** *Let $c \in \mathbb{Z}_+^k$ and define $s \in \{0,1\}^k$ entrywise where for each $i \in [k]$, $s_i = \mathbb{I}\{c_i \ge 1\}$. Then we have $[L(-Tc)]_y \ge [L(-Ts)]_y$ for all $y \in [k]$.*

*Proof.* By definition, we have

$$[L(-Tc)]_y - [L(-Ts)]_y$$
$$= \sum_{i \in [k]: i \neq y} h([-Tc]_y - [-Tc]_i) - h([-Ts]_y - [-Ts]_i)$$
$$= \sum_{i \in [k]: i \neq y} h([Tc]_i - [Tc]_y) - h([Ts]_i - [Ts]_y)$$

It suffices to show that $h([Tc]_i - [Tc]_y) - h([Ts]_i - [Ts]_y) \ge 0$ for all $i \in [k]$ such that $i \neq y$.

First, consider when $i > y$. We have

$$[Tc]_i - [Tc]_y = \sum_{j=y+1}^{i} c_j$$

Similarly, we have

$$[Ts]_i - [Ts]_y = \sum_{j=y+1}^{i} s_j = \sum_{j=y+1}^{i} \mathbb{I}\{c_j \ge 1\}.$$

From this, we see that

$$[Ts]_i - [Ts]_y \ge 1 \implies [Tc]_i - [Tc]_y \ge 1$$
$$[Ts]_i - [Ts]_y = 0 \implies [Tc]_i - [Tc]_y = 0.$$

For $i > y$, we have $h([Ts]_i - [Ts]_y) = h([Tc]_i - [Tc]_y)$.

Next, let $i < y$. We have

$$[Tc]_i - [Tc]_y = \sum_{j=i+1}^{y} -c_j.$$

Similarly, we have

$$[Ts]_i - [Ts]_y = \sum_{j=i+1}^{y} -\mathbb{I}\{c_j \ge 1\}.$$

Since $c_j \ge \mathbb{I}\{c_j \ge 1\}$, we have $[Ts]_i - [Ts]_y \ge [Tc]_i - [Tc]_y$ which implies that $h([Ts]_i - [Ts]_y) \le h([Tc]_i - [Tc]_y)$. □

**Definition S3.13.** Let $v = (v_1, \ldots, v_k) \in \mathbb{R}^k$. Define the linear map $\pi : \mathbb{R}^k \to \mathbb{R}^{k-1}$

$$\pi(v) = (v_1 - v_2, v_1 - v_3, \ldots, v_1 - v_k).$$

We observe that for each $i \in [k-1]$, we have

$$[\pi v]_i = v_1 - v_{i+1}.$$

**Definition S3.14.** Given $k \geq 2$, define the following $(k-1)$-by-$(k-1)$ square matrices $\rho_1, \rho_2, \ldots, \rho_k \in \mathbb{R}^{(k-1) \times (k-1)}$:

1. $\rho_1$ is the identity,

2. Let $z = (z_1, \ldots, z_{k-1}) \in \mathbb{R}^{k-1}$ be a vector. For each $i > 1$, define $\rho_i(z) \in \mathbb{R}^{k-1}$ entrywise for each $j \in [k-1]$ by

$$[\rho_i(z)]_j = \begin{cases} z_j - z_{i-1} & : j \neq i - 1 \\ -z_{i-1} & : j = i - 1. \end{cases} \tag{S8}$$

**Lemma S3.15** (Commuting relations). *For all $i \in [k]$, we have $\pi\sigma_i = \rho_i\pi$.*

*Proof.* If $i = 1$, then $\sigma_i$ and $\rho_i$ are both identity matrices and there is nothing to show. Otherwise, suppose that $i > 1$. Consider $v \in \mathbb{R}^k$. We first calculate $\pi\sigma_i v$. For each $j \in [k-1]$, we have

$$[\pi\sigma_i v]_j = [\sigma_i v]_1 - [\sigma_i v]_{j+1} = v_i - v_{\sigma_i(j+1)} = \begin{cases} v_i - v_{j+1} & : i \neq j + 1 \\ v_i - v_1 & : i = j + 1. \end{cases} \tag{S9}$$

Now, we compute $\rho_i\pi v$. Likewise, for each $j \in [k-1]$,

$$[\rho_i \pi v]_j = \begin{cases} [\pi v]_j - [\pi v]_{i-1} & : j \neq i - 1 \\ -[\pi v]_{i-1} & : j = i - 1. \end{cases}$$

Consider the two cases above separately: for $j \neq i - 1$, we have

$$[\pi v]_j - [\pi v]_{i-1} = (v_1 - v_{j+1}) - (v_1 - v_i) = v_i - v_{j+1}.$$

On the other hand, for $i = j + 1$, we have

$$-[\pi v]_{i-1} = -(v_1 - v_i) = v_i - v_1.$$

Thus, we have $[\pi\sigma_i v]_j = [\rho_i\pi v]_j$ for all $j$ which implies that $\pi\sigma_i v = \rho_i\pi v$. Since $v$ was arbitrary, we have $\pi\sigma_i = \rho_i\pi$. $\qquad\square$

**Definition S3.16.** The *reduced WW hinge function* $H : \mathbb{R}^{k-1} \to \mathbb{R}_{\geq 0}$ is defined as

$$H(z) = \sum_{i=1}^{k-1} h(z_i).$$

**Definition S3.17.** For $z \in \mathbb{R}^{k-1}$, the *reduced WW hinge loss* $\L(z) \in \mathbb{R}^k$ is defined entrywise for each $y \in [k]$ by

$$[\L(z)]_y = H(\rho_y z).$$

**Lemma S3.18.** *For all $v \in \mathbb{R}^k$, we have $\L(\pi v) = L(v)$.*

*Proof.* We first check for all $y \in [k]$ that

$$\sum_{i \in [k] \,:\, i \neq y} h(v_y - v_i) = H(\pi\sigma_i v). \tag{S10}$$

Unpacking the definition, we have $H(\pi\sigma_y v) = \sum_{i \in [k-1]} h([\pi\sigma_y v]_i)$. Now, if $y = 1$, then $[\pi v]_i = v_1 - v_{i+1}$ for all $i \in [k-1]$. Hence, eq. (S10) holds. If $y > 1$. Then eq. (S10) follows from the

expression for $[\pi\sigma_y v]_i$ computed in eq. (S9). Thus, we have proven eq. (S10) for all $y \in [k]$. To conclude, we have

$$[L(v)]_y = \sum_{i \in [k] : i \neq y} h(v_y - v_i) \tag{S11}$$

$$= H(\pi\sigma_y v) \tag{S12}$$

$$= H(\rho_y \pi v) \tag{S13}$$

$$= [Ł(\pi v)]_y \tag{S14}$$

where in eq. (S13), we applied Lemma S3.15. $\qquad\square$

**Lemma S3.19.** *Let* $n \in [k-1]$. *If* $p \in \Delta_\downarrow^k$, *then*

$$\underline{L}^n(p) = \min_{v \in \mathcal{C}_\mathbb{Z} : v_n = 0} \langle p, L(v) \rangle.$$

*Proof.* Define

$$\mathcal{N}^n = \{v \in \mathbb{R}^k : v_1 = \cdots = v_n = 0, \ v_i \leq 0, \ \forall i \in [k]\}.$$

We first claim that

$$\underline{L}^n(p) = \inf_{v \in \mathcal{N}^n} \langle p, L(v) \rangle. \tag{S15}$$

Since $\mathcal{N}^n \subseteq \{v \in \mathbb{R}^k : |\arg\max v| \geq n\}$, the "$\leq$" part of eq. (S15) is obvious. For the "$\geq$" part, let $v \in \mathbb{R}^k$ be such that $|\arg\max v| \geq n$. Then $w = v - \mathbf{1}^k \max_{i \in [k]} v_i$ is such that $w \in \mathcal{N}^n$. Furthermore, by Lemma S3.1, we have $\langle p, L(v) \rangle = \langle p, L(w) \rangle$. Thus, we have proven the claim.

Next, observe that if $v \in \mathcal{N}^n$, then

$$[\pi v]_i = v_1 - v_{i+1} \begin{cases} = 0 & : i \leq n - 1 \\ \geq 0 & : i \geq n. \end{cases}$$

Therefore, we have

$$\pi(\mathcal{N}^n) = \{z \in \mathbb{R}^{k-1} : z \geq 0, \ z_i = 0, \forall i \in [n-1]\}$$

where $[0] = \emptyset$. Introducing the change of variable $z = \pi v \in \mathbb{R}^{k-1}$, we have

$$\inf_{v \in \mathcal{N}^n} \langle p, L(v) \rangle = \inf_{v \in \mathcal{N}^n} \langle p, Ł(\pi v) \rangle \quad \because \text{Lemma S3.18} \tag{S16}$$

$$= \inf_{z \in \pi(\mathcal{N})} \langle p, Ł(z) \rangle \tag{S17}$$

$$= \inf_{\substack{z \in \mathbb{R}^{k-1} : z \geq 0 \\ z_i = 0, \ \forall i \in [n-1]}} \langle p, Ł(z) \rangle \tag{S18}$$

Below, let $\mathbf{1} := \mathbf{1}^{k-1}$. Unwinding the definition, we have

$$\langle p, Ł(z) \rangle = \sum_{i \in [k]} p_i H(\rho_i z) = \sum_{i \in [k]} p_i \mathbf{1}' \left[\mathbf{1} - \rho_i z\right]_+.$$

Using slack variables $\xi_i \geq \left[\mathbf{1} - \rho_i z\right]_+$, we can rewrite eq. (S18) as the following linear program:

$$\min_{z \in \mathbb{R}^{k-1}} \min_{(\xi_1, \dots, \xi_k) : \xi_i \in \mathbb{R}^{k-1}} \sum_i p_i \mathbf{1}' \xi_i \tag{S19}$$

$$s.t. \quad \xi_i \geq \mathbf{1} - \rho_i z \tag{S20}$$

$$\xi_i \geq 0, \quad \forall i \in [k] \tag{S21}$$

$$z \geq 0, \tag{S22}$$

$$z_i = 0, \forall i \in [n-1]. \tag{S23}$$

By Bertsimas and Tsitsiklis [1, Corollary 3.2], for a linear programming minimization problem over a nonempty polyhedron, one of the following must be true: 1) the optimal cost is $-\infty$ or 2) a feasible

minimum exists. Since eq. (S19) is nonnegative and the feasible region is nonempty, a feasible minimum exists. Let

$$R = \begin{bmatrix} \rho_1 \\ \rho_2 \\ \vdots \\ \rho_k \end{bmatrix} \in \mathbb{R}^{k(k-1)\times(k-1)}, \quad X = \begin{bmatrix} \xi_1 \\ \xi_2 \\ \vdots \\ \xi_k \end{bmatrix} \in \mathbb{R}^{k(k-1)}, \quad p \otimes \mathbf{1} = \begin{bmatrix} p_1\mathbf{1} \\ p_2\mathbf{1} \\ \vdots \\ p_k\mathbf{1} \end{bmatrix} \in \mathbb{R}^{k(k-1)}.$$

We claim that

$$\underline{L}^n(p) = \min_{z\in\mathbb{R}_+^{k-1}:z_i=0\,\forall i\in[n-1]} \langle p, \L(z) \rangle. \tag{S24}$$

We first consider the case when $n=1$ where we have $\underline{L}^1 = \underline{L}$. In this case, the linear program eq. (S19) can be rewritten as

$$\underline{L}(p) = \min_{z\in\mathbb{R}^{k-1}} \min_{X\in\mathbb{R}^{k(k-1)}} \quad (p\otimes\mathbf{1})'X$$
$$s.t. \quad X + Rz \geq \mathbf{1}$$
$$X \geq 0$$
$$z \geq 0.$$

For a positive integer $m$, let $I_m$ denote the $m\times m$ identity matrix. Thus,

$$\min_{z\in\mathbb{R}^{k-1},\,X\in\mathbb{R}^{k(k-1)}} \quad (p\otimes\mathbf{1})'X \tag{S25}$$

$$s.t. \quad \underbrace{\begin{bmatrix} R & I_{k(k-1)} \\ I_{k-1} & 0 \\ 0 & I_{k(k-1)} \end{bmatrix}}_{=:A} \begin{bmatrix} z \\ X \end{bmatrix} \geq \begin{bmatrix} \mathbf{1} \\ 0 \\ 0 \end{bmatrix}. \tag{S26}$$

We prove that $A$ is totally unimodular (TUM). The matrix $R$ has the property that every row has at most one 1 and at most one $-1$, with all other entries being zeros. Hence, $R$ is TUM by the Hoffman's sufficient condition Lawler [2]. Thus, (horizontally) concatenating $R$ with an identity matrix, i.e., $R_0 := \begin{bmatrix} R & I_{k(k-1)} \end{bmatrix}$ results in another TUM matrix $R_0$. Finally, $A$ is the (vertical) concatenation of $R_0$ with another identity matrix, i.e., $A = \begin{bmatrix} R_0 \\ I_{k(k-1)} \end{bmatrix}$. Hence, $A$ is also TUM.

By a well-known result in combinatorial optimization Lawler [2], there exists an integral solution $(X^*, z^*)$ to eq. (S25). In particular, $z^* \in \mathbb{Z}_+^{k-1}$. Thus, we have proven that

$$\underline{L}(p) = \langle p, \L(z^*) \rangle = \min_{z\in\mathbb{Z}_+^{k-1}} \langle p, \L(z) \rangle.$$

This proves eq. (S24) for the case when $n=1$. For $n>1$, we define the matrix $J \in \mathbb{R}^{(n-1)\times(k-1)}$ to be the first $n-1$ rows of the $(k-1)$-by-$(k-1)$ identity matrix. In other words, for $i\in[n-1]$ and $j\in[k-1]$,

$$J_{ij} = \begin{cases} 1 & : i=j \\ 0 & : i\neq j \end{cases}.$$

Thus, we have

$$\underline{L}^n(p) = \min_{z\in\mathbb{R}^{k-1},\,X\in\mathbb{R}^{k(k-1)}} \quad (p\otimes\mathbf{1})'X$$

$$s.t. \quad \underbrace{\begin{bmatrix} R & I_{k(k-1)} \\ I_{k-1} & 0 \\ 0 & I_{k(k-1)} \\ -J & 0 \end{bmatrix}}_{=:B} \begin{bmatrix} z \\ X \end{bmatrix} \geq \begin{bmatrix} \mathbf{1} \\ 0 \\ 0 \\ 0 \end{bmatrix}.$$

The matrix $B$ is formed by duplicating rows of $A$ and multiplying the duplicated row by $-1$. Thus, $B$ is also TUM. This proves eq. (S24).

Below, let $z^*$ be a solution to eq. (S24). Define $v^* = \begin{pmatrix} 0 \\ -z^* \end{pmatrix}$. Furthermore, $\pi(v^*) = z^*$ and so

$$
\begin{aligned}
\underline{L}^n(p) &= \langle p, \mathcal{L}(z^*) \rangle \\
&= \langle p, \mathcal{L}(\pi(v^*)) \rangle \\
&= \langle p, L(v^*) \rangle.
\end{aligned}
$$

Pick $\sigma \in \mathfrak{S}_k$ such that $\sigma v^* \in \mathbb{R}^k_\downarrow$. First we note that $L(\sigma v^*) \in \mathbb{R}^k_\uparrow$ by Lemma S3.6. Next, by Corollary S3.5, $L(\sigma v^*) = \sigma L(v^*)$. Hence, by Lemma S3.8

$$
\langle p, L(v^*) \rangle \geq \langle p, \sigma L(v^*) \rangle = \langle p, L(\sigma v^*) \rangle
$$

which implies that $\sigma v^*$ is optimal. Also, we observe that $\sigma v^* \in \mathcal{M}_{\mathbb{Z}}$. By Lemma S3.11, we can write $\sigma v^* = -Tc$ for some $c \in \mathbb{Z}^k_+$. Note that since $z_1^* = \cdots = z_{n-1}^* = 0$, the vector $v^*$ has at least $n$ entries equal to 0. Since $v^* \leq 0$, we must have that $v_1 = \cdots = v_n^* = 0$. Thus, $c_1 = \cdots c_n = 0$ as well. Let $s \in \{0,1\}^k$ be as defined in Lemma S3.12. Then we have

$$
\underline{L}^n(p) \geq \langle p, L(\sigma v^*) \rangle = \langle p, L(-Tc) \rangle \geq \langle p, L(-Ts) \rangle.
$$

Hence, we have $\underline{L}(p) = \langle p, L(-Ts) \rangle$. Since $s_i = \mathbb{I}\{c_i \geq 1\}$, we have $s_1 = \cdots = s_n = 0$ which implies that $[-Ts]_1 = \cdots = [-Ts]_n = 0$. Consider the case when there exists some $i \in \{n+1, \ldots, k\}$ such that $s_i = 1$, then we have $-Ts \in \mathcal{C}_{\mathbb{Z}}$ which completes the proof of Lemma S3.19. Now, consider the case where there does not exists such $i$. Then we must have $s = 0$ and also $-Ts = 0$. Therefore, we have $\underline{L}^n(p) = \langle p, L(0) \rangle$. Define $\tilde{v} \in \mathbb{R}^k$ entrywise by

$$
[\tilde{v}]_i = \begin{cases} 0 & : i \neq k \\ -1 & : i = k \end{cases}
$$

Noting that $k \in \arg\min_{i \in [k]} p_i$ by the assumption that $p \in \Delta^k_\downarrow$. By Lemma S3.20 below, we get that $\langle p, L(\tilde{v}) \rangle \leq \langle p, L(0) \rangle$ which implies that $\langle p, L(\tilde{v}) \rangle = \underline{L}^n(v)$. Clearly, $\tilde{v} \in \mathcal{C}_{\mathbb{Z}}$ and $\tilde{v}_n = 0$, which implies that $\tilde{v}$ is feasible for the optimization in Lemma S3.19. $\qquad \square$

**Lemma S3.20.** *Let $p \in \Delta^k$ and $i^* \in \arg\min_{i \in [k]} p_i$. Consider the vector $\tilde{v} \in \mathbb{R}^k$ defined by*

$$
[\tilde{v}]_i = \begin{cases} 0 & : i \neq i^* \\ -1 & : i = i^* \end{cases}
$$

*Then*

1. *$p_{i^*} \leq \frac{1}{k}$*

2. *$p_i = \frac{1}{k}$ for all $i$ if and only if $p_{i^*} = \frac{1}{k}$*

3. *$\langle p, L(0) \rangle \geq \langle p, L(\tilde{v}) \rangle$ with equality if and only if $p_{i^*} = \frac{1}{k}$.*

*Proof.* If $p_{i^*} > \frac{1}{k}$, then we would have $\sum_i p_i \geq k p_{i^*} > 1$, a contradiction. This proves that $p_{i^*} \leq \frac{1}{k}$. For the second item, the "only if" direction is obvious. For the "if" direction, note that if $p_i > \frac{1}{k}$ for any $i$, then we again obtain $\sum_i p_i > 1$, a contradiction. For the third item, first observe that

$$
[L(0)]_i = \sum_{j \in [k]: j \neq i} h(0) = k - 1.
$$

Thus, $L(0) = (k-1)\mathbf{1}^k$ and $\langle p, L(0) \rangle = k - 1$. Next, we only $L(\tilde{v})$. For $i \neq i^*$, we have

$$
[L(\tilde{v})]_i = \sum_{j \in [k]: j \neq i} h(\tilde{v}_i - \tilde{v}_j) = h(1) + \sum_{j \in [k]: j \neq i, j \neq i^*} h(0) = k - 2.
$$

When $i = i^*$, we have

$$
[L(\tilde{v})]_{i^*} = \sum_{j \in [k]: j \neq i^*} h(\tilde{v}_{i^*} - \tilde{v}_j) = \sum_{j \in [k]: j \neq i^*} h(-1) = 2(k-1) = k - 2 + k.
$$

From this, we deduce that

$$
\langle p, L(\tilde{v}) \rangle = k - 2 + k p_{i^*}.
$$

Therefore, we have $p_{i^*} \leq \frac{1}{k}$ and so

$$
\langle p, L(\tilde{v}) \rangle = k - 2 + k p_{i^*} \leq k - 2 + 1 = k - 1 = \langle p, L(0) \rangle.
$$

Note if if $p_{i^*} < \frac{1}{k}$, then the inequality above is strict. $\qquad \square$

## S3.1 Proof of Theorem 3.7

*Proof of Theorem 3.7.* Recall that $\underline{L}(p) = \min_{v \in \mathbb{R}^k} \langle p, L(v) \rangle$. Since $\mathbb{R}^k \supseteq \mathfrak{S}_k \mathcal{C}_{\mathbb{Z}}$, we immediately have $\underline{L}(p) \leq \min_{v \in \mathfrak{S}_k \mathcal{C}_{\mathbb{Z}}} \langle p, L(v) \rangle$. Below, we focus on the other inequality.

Pick $\sigma \in \mathfrak{S}_k$ such that $\sigma p \in \Delta_{\downarrow}^k$. By Lemma S3.19 where $n = 1$, we have

$$\underline{L}(\sigma p) = \min_{v \in \mathcal{C}_{\mathbb{Z}}} \langle \sigma p, L(v) \rangle.$$

Now, by Corollary S3.5, we have

$$\langle \sigma p, L(v) \rangle = \langle p, \sigma' L(v) \rangle = \langle p, L(\sigma' v) \rangle.$$

Thus,

$$
\begin{aligned}
\underline{L}(p) = \underline{L}(\sigma p) \quad & \because \text{Lemma S3.7} \\
= \min_{v \in \mathcal{C}_{\mathbb{Z}}} \langle p, L(\sigma' v) \rangle & \\
= \min_{v \in \sigma' \mathcal{C}_{\mathbb{Z}}} \langle p, L(v) \rangle \quad & \because \text{change of variables} \\
\geq \min_{v \in \mathfrak{S}_k \mathcal{C}_{\mathbb{Z}}} \langle p, L(v) \rangle &
\end{aligned}
$$

where for the last equality, we used the fact that $\sigma' \mathcal{C}_{\mathbb{Z}} \subseteq \mathfrak{S}_k \mathcal{C}_{\mathbb{Z}}$. $\qquad\square$

**Lemma S3.21.** *Let $s \in \{0, 1\}^k$ be such that $s_1 = 0$. Then*

$$[DL(-Ts)]_y = \begin{cases} \min\{i \in [k] : s_i = 1\} - 2 & : y = 1 \\ \min\{i \in [k] : s_i = 1, i > y\} - 1 & : s_y = 1, y > 1 \\ 0 & : s_y = 0, y > 1 \end{cases} \tag{S27}$$

*Proof.* By the definition of $T$, we have

$$[Ts]_j = \sum_{i=1}^{j} s_i. \tag{S28}$$

First, consider the case when $y = 1$. Then by eq. (S28) we have $[-Ts]_1 = 0$. Furthermore,

$$
\begin{aligned}
[DL(-Ts)]_1 = [L(-Ts)]_1 & \\
= \sum_{i \in [k]: i \neq 1} h([-Ts]_1 - [-Ts]_i) & \\
= \sum_{i \in [k]: i \neq 1} h([Ts]_i) &
\end{aligned}
$$

Note that by eq. (S28), we have $[Ts]_i \geq 1$ if $i \geq \min\{j : s_j = 1\}$ and $[Ts]_i = 0$ otherwise. Hence, we get

$$
\begin{aligned}
[DL(-Ts)]_1 &= \sum_{i \in [k]: 1 < i < \min\{j: s_j = 1\}} h([Ts]_i) \\
&= \sum_{i \in [k]: 1 < i < \min\{j: s_j = 1\}} 1 \\
&= \min\{j \in [k] : s_j = 1\} - 2.
\end{aligned}
$$

This proves the first case of eq. (S27). Below, let $y > 1$. We have

$$[DL(-Ts)]_y \tag{S29}$$

$$= \sum_{i \in [k]: i \neq y} h([-Ts]_y - [-Ts]_i) - \sum_{i \in [k]: i \neq y-1} h([-Ts]_{y-1} - [-Ts]_i) \tag{S30}$$

$$= \sum_{i \in [k]: i \neq y} h([Ts]_i - [Ts]_y) - \sum_{i \in [k]: i \neq y-1} h([Ts]_i - [Ts]_{y-1}) \tag{S31}$$

$$= \sum_{i \in [k]: i < y-1} h([Ts]_i - [Ts]_y) - h([Ts]_i - [Ts]_{y-1}) \tag{S32}$$

$$+ h([Ts]_{y-1} - [Ts]_y) - h([Ts]_y - [Ts]_{y-1}) \tag{S33}$$

$$+ \sum_{i \in [k]: i > y} h([Ts]_i - [Ts]_y) - h([Ts]_i - [Ts]_{y-1}) \tag{S34}$$

If $s_y = 0$, then $[Ts]_y = [Ts]_{y-1}$ and so we have $[DL(-Ts)]_y = 0$. This proves the last case of eq. (S27).

Below, assume the setting of the second case, i.e., $y > 1$ and $s_y = 1$. We first evaluate eq. (S32). Since $i < y - 1$, we have

$$([Ts]_i - [Ts]_y) - ([Ts]_i - [Ts]_{y-1}) = [Ts]_{y-1} - [Ts]_y = -1$$

and

$$([Ts]_i - [Ts]_{y-1}) \leq 0.$$

The two preceding facts together imply that

$$h([Ts]_i - [Ts]_y) - h([Ts]_i - [Ts]_{y-1}) = 1$$

and so

$$\sum_{i \in [k]: i < y-1} h([Ts]_i - [Ts]_y) - h([Ts]_i - [Ts]_{y-1}) = y - 2.$$

Next, we evaluate eq. (S33)

$$h([Ts]_{y-1} - [Ts]_y) - h([Ts]_y - [Ts]_{y-1}) = h(-1) - h(1) = 2.$$

Finally, we evaluate eq. (S34). Since $i > y$, we have

$$[Ts]_i - [Ts]_y = \sum_{j=y+1}^{i} s_i.$$

From this, we see that

$$[Ts]_i - [Ts]_y \begin{cases} = 0 & : i < \min\{j \in [k] : j > y, \, s_j = 1\} \\ \geq 1 & : \text{otherwise.} \end{cases}$$

Hence,

$$h([Ts]_i - [Ts]_y) \begin{cases} = 1 & : i < \min\{j \in [k] : j > y, \, s_j = 1\} \\ = 0 & : \text{otherwise.} \end{cases}$$

On the other hand, $[Ts]_i - [Ts]_{y-1} = \sum_{j=y}^{i} s_i \geq s_y = 1$ and so $h([Ts]_i - [Ts]_{y-1}) = 0$. Therefore,

$$\sum_{i \in [k]: i > y} h([Ts]_i - [Ts]_y) - h([Ts]_i - [Ts]_{y-1})$$

$$= \min\{j \in [k] : j > y, \, s_j = 1\} - y - 1$$

Putting it all together, we have

$$[DL(-Ts)]_y = y - 2 + 2 + \min\{j \in [k] : j > y, \, s_j = 1\} - y - 1$$
$$= \min\{j \in [k] : j > y, \, s_j = 1\} - 1.$$

$\square$

## S3.2   Proof of Theorem 3.8

*Proof of Theorem 3.8.* Let $\mathbf{S} = (S_1, \ldots, S_l) \in \mathcal{OP}_k$. Pick $\sigma$ such that $\sigma\varphi(\mathbf{S})$ is monotonic non-increasing. Hence, we have

$$\sigma\varphi(\mathbf{S}) = -[\,\underbrace{0, \ldots, 0}_{|S_1|\text{-times}}\,,\,\underbrace{1, \ldots, 1}_{|S_2|\text{-times}}\,, \ldots, \underbrace{l-1, \ldots, l-1}_{|S_l|\text{-times}}].$$

For each $i = 1, \ldots, l-1$, define $c_i(\mathbf{S}) = |S_1| + \cdots + |S_i|$.

Note that

$$S_1 \cup \cdots \cup S_i = \{j \in [k] : 0 \geq [\varphi(\mathbf{S})]_j \geq -(i-1)\} \tag{S35}$$
$$= \{\sigma(1), \sigma(2), \ldots, \sigma(c_i(\mathbf{S}))\}. \tag{S36}$$

Also, note that by definition, $c_i(\mathbf{S})$ is precisely the index in $[k-1]$ such that

$$\begin{cases} [\sigma\varphi(\mathbf{S})]_{c_i(\mathbf{S})} = -(i-1) \\ [\sigma\varphi(\mathbf{S})]_{c_i(\mathbf{S})+1} = -i. \end{cases}$$

Motivated by this, we define $\zeta(\mathbf{S}) \in \{0,1\}^k$ where

$$[\zeta(\mathbf{S})]_j = \begin{cases} 1 & : j = c_i(\mathbf{S}) + 1 \text{ for some } i = 1, \ldots, l-1 \\ 0 & : \text{otherwise.} \end{cases}$$

Then

$$\sigma\varphi(\mathbf{S}) = -T\zeta(\mathbf{S}). \tag{S37}$$

Next, note that

$$\langle p, L(\varphi(\mathbf{S})) \rangle = \langle p, L(\sigma'\sigma\varphi(\mathbf{S})) \rangle \tag{S38}$$
$$= \langle p, \sigma' L(\sigma\varphi(\mathbf{S})) \rangle \tag{S39}$$
$$= \langle \sigma p, L(\sigma\varphi(\mathbf{S})) \rangle \tag{S40}$$
$$= \langle T'(\sigma p), DL(\sigma\varphi(\mathbf{S})) \rangle \tag{S41}$$
$$= \langle T'(\sigma p), DL(-T\zeta(\mathbf{S})) \rangle \tag{S42}$$

where eq. (S38) is by $\sigma' = \sigma^{-1}$, eq. (S39) is by Corollary S3.5, eq. (S40) is a basic property of the dot product, eq. (S41) is by Lemma S3.9, eq. (S42) is by eq. (S37).

We first calculate $DL(-T\zeta(\mathbf{S}))$ by applying eq. (S27) from Lemma S3.21 to $s = \zeta(\mathbf{S})$. For the case $y = 1$ of eq. (S27), we have

$$[DL(-T\zeta(\mathbf{S}))]_1 = \min\{j \in [k-1] : [\zeta(\mathbf{S})]_j = 1\} - 2$$
$$= c_1(\mathbf{S}) + 1 - 2$$
$$= |S_1| - 1.$$

By definition, for $y > 1$, we note that $[\zeta(\mathbf{S})]_y = 1$ if and only if $y = c_i(\mathbf{S}) + 1$ for some $i \in \{1, \ldots, l-1\}$. Thus,

$$[DL(-T\zeta(\mathbf{S}))]_{c_i(\mathbf{S})+1} = \min\{j \in [k] : [\zeta(\mathbf{S})]_j = 1, j > c_i(\mathbf{S}) + 1\} - 1$$
$$= (c_{i+1}(\mathbf{S}) + 1) - 1 = c_{i+1}(\mathbf{S}).$$

We summarize the above as follows:

$$[DL(-T\zeta(\mathbf{S}))]_y = \begin{cases} |S_1| - 1 & : y = 1 \\ c_{i+1}(\mathbf{S}) & : y = c_i(\mathbf{S}) + 1 \text{ for some } i \in [l-1] \\ 0 & : \text{otherwise.} \end{cases}$$

Next, we calculate $T'(\sigma p)$. Note that

$$[T'(\sigma p)]_y = p_{\sigma(y)} + p_{\sigma(y+1)} + \cdots + p_{\sigma(k)}$$
$$= 1 - \left(p_{\sigma(1)} + \cdots + p_{\sigma(y-1)}\right).$$

In particular, $[T'(\sigma p)]_1 = 1$. Hence,

$$
\begin{aligned}
\langle p, L(\varphi(\mathbf{S}))\rangle &\\
&= \langle T'(\sigma p), DL(-T\zeta(\mathbf{S}))\rangle \\
&= [T'(\sigma p)]_1(|S_1| - 1) \\
&\quad + \sum_{i=1}^{l-1} \left([T'(\sigma p)]_{c_i(\mathbf{S})+1}\right) c_{i+1}(\mathbf{S}) \\
&= |S_1| - 1 \\
&\quad + \sum_{i=1}^{l-1} \left(1 - \left(p_{\sigma(1)} + \cdots + p_{\sigma(c_i(\mathbf{S}))}\right)\right) c_{i+1}(\mathbf{S}).
\end{aligned}
$$

Recall from eq. (S36)

$$
\{\sigma(1), \sigma(2), \ldots, \sigma(c_i(\mathbf{S}))\} = S_1 \cup \cdots \cup S_i.
$$

Hence,

$$
\left(1 - \left(p_{\sigma(1)} + \cdots + p_{\sigma(c_i(\mathbf{S}))}\right)\right) = \Pr_{Y \sim p}(Y \notin S_1 \cup \cdots \cup S_i).
$$

Putting it all together, we have

$$
\begin{aligned}
\langle p, L(\varphi(\mathbf{S}))\rangle &= |S_1| - 1 + \sum_{i=1}^{l_{\mathbf{S}}-1} |S_1 \cup \cdots \cup S_{i+1}| \Pr_{Y \sim p}(Y \notin S_1 \cup \cdots \cup S_i) \\
&= \mathbb{E}_{Y \sim p}\left[[\ell(\mathbf{S})]_Y\right] \\
&= \langle p, \ell(\mathbf{S})\rangle
\end{aligned}
$$

This concludes the proof of Theorem 3.8. $\qquad\square$

## S4 Minimally emblematic losses

We first introduce some basic properties of hyperplane arrangements that will be needed later.

**Definition S4.1.** A *hyperplane* in $\mathbb{R}^d$ is a subset $H \subseteq \mathbb{R}^d$ of the form $H = \{v \in \mathbb{R}^k : b - \langle a, v\rangle = 0\}$ for some (column) vector $a \in \mathbb{R}^k$ and $b \in \mathbb{R}$.

**Definition S4.2.** Define the following:

1. A *hyperplane arrangement* is a set of hyperplanes $\{H_n\}_{n \in I}$ indexed by a finite set $I$. Let the hyperplanes be written as $H_n = \{v \in \mathbb{R}^k : b^{(n)} - \langle a^{(n)}, v\rangle = 0\}$ for each $n \in I$.

2. Define $\mathfrak{s} : \mathbb{R}^k \to \{-1, 0, 1\}^I$ entrywise by

$$
[\mathfrak{s}(v)]_n = \mathrm{sgn}\left(b^{(n)} - \langle a^{(n)}, v\rangle\right), \quad \text{where } \forall t \in \mathbb{R}, \ \mathrm{sgn}(t) = \begin{cases} 1 & : t > 0 \\ 0 & : t = 0 \\ -1 & : t < 0 \end{cases}.
$$

3. Define the set $\Theta := \mathfrak{s}(\mathbb{R}^k) \subseteq \{-1, 0, 1\}^I$.

4. For each $\theta \in \Theta$, define

$$
\tilde{P}_\theta := \mathfrak{s}^{-1}(\theta) = \{v \in \mathbb{R}^k : \mathfrak{s}(v) = \theta\} \quad \text{and} \quad P_\theta := \mathrm{cl}(\tilde{P}_\theta)
$$

where $\mathrm{cl}$ denotes the closure of a set in $\mathbb{R}^k$ with the Euclidean topology.

**Definition S4.3.** An *affine subspace* of $\mathbb{R}^k$ is a set of the form $W + v$ where $W \subseteq \mathbb{R}^k$ is a linear subspace and $v \in \mathbb{R}^k$ is a vector. Let $C$ be a convex set. The *affine hull* $\mathrm{Aff}(C)$ of $C$ is defined as the smallest affine subspace containing $C$. The *relative interior* of $C$, denoted $\mathrm{relint}(C)$, is defined as the subset of $v \in C$ such that for all $\epsilon > 0$ sufficiently small, we have that

$$
\mathrm{Aff}(C) \cap \{w \in \mathbb{R}^k : \|w - v\| < \epsilon\} \subseteq C.
$$

In other words, $\mathrm{relint}(C)$ is an open subset of $\mathrm{Aff}(C)$. Here $\| \bullet \|$ is the Euclidean 2-norm on $\mathbb{R}^k$.

The following result is "folklore". Since we cannot find its proof, we prove it here.

**Lemma S4.4.** *Let $\{H_n\}_{n \in I}$ be an arrangement of hyperplanes. Adopt all notations from Definition S4.2. The following are true:*

*1. For all $\theta \in \Theta$, $\tilde{P}_\theta = \left\{ v \in \mathbb{R}^k : \begin{cases} \theta_n(b^{(n)} - \langle a^{(n)}, v \rangle) > 0 & : \theta_n \neq 0 \\ b^{(n)} - \langle a^{(n)}, v \rangle = 0 & : \theta_n = 0 \end{cases}, \forall n \in I \right\}$,*

*2. For all $\theta \in \Theta$, $P_\theta = \left\{ v \in \mathbb{R}^k : \begin{cases} \theta_n(b^{(n)} - \langle a^{(n)}, v \rangle) \geq 0 & : \theta_n \neq 0 \\ b^{(n)} - \langle a^{(n)}, v \rangle = 0 & : \theta_n = 0 \end{cases}, \forall n \in I \right\}$,*

*3. For all $\theta \in \Theta$, $\mathrm{relint}(P_\theta) = \tilde{P}_\theta$,*

*4. $\bigsqcup_{\theta \in \Theta} \mathrm{relint}(P_\theta) = \mathbb{R}^k$ as a disjoint union.*

*Proof.* First, we note that item 1 follows directly from definition.

For item 2, let $Q_\theta$ denote the set on the right hand side of the identity. We want to show that $P_\theta = Q_\theta$. Recall that $P_\theta = \mathrm{cl}(\tilde{P}_\theta)$ is by definition the smallest closed set containing $\tilde{P}_\theta$. Clearly, $Q_\theta$ is a closed set. Furthermore, by item 1, we have $\tilde{P}_\theta \subseteq Q_\theta$. Thus, we have the $P_\theta \subseteq Q_\theta$.

Conversely, let $v \in Q_\theta$ and $w \in \tilde{P}_\theta$. Then by item 1, we have that $(1 - \lambda)w + \lambda v \in \tilde{P}_\theta$ for all $\lambda \in [0, 1)$. Now, $\lim_{\lambda \to 1}(1 - \lambda)w + \lambda v = v$. Since $\mathrm{cl}(\tilde{P}_\theta)$ is closed, it contains all limits. Hence $v \in \mathrm{cl}(\tilde{P}_\theta) = P_\theta$, as desired. This proves that $Q_\theta \subseteq P_\theta$, as desired.

Next, we prove item 3. From the first paragraph of Ben-Tal and Nemirovski [3, Section 1.1.6.D], we have $\mathrm{relint}(\tilde{P}_\theta) \subseteq \tilde{P}_\theta \subseteq \mathrm{cl}(\tilde{P}_\theta)$. By Ben-Tal and Nemirovski [3, Theorem 1.1.1 (iv)], we have $\mathrm{relint}(\tilde{P}_\theta) = \mathrm{relint}(\mathrm{cl}(\tilde{P}_\theta))$. By definition $P_\theta = \mathrm{cl}(\tilde{P}_\theta)$. Putting it all together, we get $\mathrm{relint}(P_\theta) \subseteq \tilde{P}_\theta$.

For the other inclusion, let $v \in \tilde{P}_\theta$. Let

$$W = \{ v \in \mathbb{R}^k : b^{(n)} - \langle a^{(n)}, v \rangle = 0, \forall n \in I \text{ such that } \theta_n = 0 \}.$$

Then by item 2, $W$ is an affine subspace containing $P_\theta$. Thus, by definition of the affine hull, we have $W \supseteq \mathrm{Aff}(P_\theta)$. Furthermore, by item 1, we have, for all $\epsilon > 0$ sufficiently small, that $W \cap \{ w \in \mathbb{R}^k : \|w - v\| < \epsilon \} \subseteq P_\theta$. This proves that $v \in \mathrm{relint}(P_\theta)$ and so $\tilde{P}_\theta \subseteq \mathrm{relint}(P_\theta)$.

Finally, we prove item 4

$$\bigsqcup_{\theta \in \Theta} \mathrm{relint}(P_\theta) = \bigsqcup_{\theta \in \Theta} \tilde{P}_\theta = \bigsqcup_{\theta \in \mathfrak{s}(\mathbb{R}^k)} \mathfrak{s}^{-1}(\theta) = \mathbb{R}^k,$$

where for the middle equality, we recall that $\Theta = \mathfrak{s}(\mathbb{R}^k)$ by definition. $\qquad \square$

## S4.1 Semiordered hyperplane arrangement

Below, we apply the results of Lemma S4.4 to the "semiorder hyperplane arrangement", which is closely connected to the WW-hinge loss.

**Definition S4.5.** The *semiorder hyperplane arrangement* is the hyperplane arrangement in $\mathbb{R}^k$ indexed by the finite set $I = \{(i, j) \in [k] \times [k] : i \neq j\}$ with the $(i, j)$-th hyperplane given by $H_{(i,j)} = \{v \in \mathbb{R}^k : 1 - (v_i - v_j) = 0\}$.

**Lemma S4.6.** *Let $L : \mathbb{R}^k \to \mathbb{R}^k_+$ be the WW-hinge loss and $\mathfrak{S}_k \mathcal{C}_{\mathbb{Z}}$ be as in Definition 3.3. Let $\{H_{(i,j)}\}_{(i,j) \in I}$ be the semiorder hyperplane arrangement as in Definition S4.5. Adopt all notations from Definition S4.2. Then we have for all $\theta \in \Theta$ that*

*1. the restriction of $L$ to $P_\theta$, denoted $L|_{P_\theta}$, is an affine function,*

*2. $P_\theta \cap \mathfrak{S}_k \mathcal{C}_{\mathbb{Z}}$ is nonempty.*

*Proof.* For the first item, fix some $i \in [k]$ and note that

$$[L(v)]_i = \sum_{j \in [k]: j \neq i} \max\{0, 1 - (v_i - v_j)\}.$$

Fix $(i, j) \in I$ where $I$ is as in Definition S4.5. Then by Lemma S4.4 item 2, for all $v \in P_\theta$, we have

$$\max\{0, 1 - (v_i - v_j)\} = \begin{cases} 1 - (v_i - v_j) & : \theta_{(i,j)} = 1 \\ 0 & : \text{otherwise.} \end{cases}$$

In either case, $\max\{0, 1 - (v_i - v_j)\}$ is affine over $P_\theta$.

Next, we prove the second item. Define $H_0 = \{v \in \mathbb{R}^k : \sum_{i \in [k]} v_i = 0\}$. Then $H_0 \cap P_\theta$ is nonempty for all $\theta \in \Theta$. To see this, first note that $P_\theta$ is nonempty by construction. Furthermore, if $v \in P_\theta$ then $v + c\mathbf{1}^k \in P_\theta$ as well for any $c \in \mathbb{R}$. Thus, $v + (-(1/k) \sum_{i \in [k]} v_i)\mathbf{1}^k \in H_0 \cap P_\theta$.

**Lemma S4.7.** $H_0 \cap P_\theta$ *does not contain any line.*

*Proof.* Suppose that this is false, i.e., $\mathfrak{l} \subseteq H_0 \cap P_\theta$ where $\mathfrak{l} \subseteq \mathbb{R}^k$ is a line. In particular, $\mathfrak{l} \subseteq H_0$. This means that $\mathfrak{l} = \{cw : c \in \mathbb{R}\}$ where $w \in H_0$ is a nonzero vector. Thus, there exists $i \neq j$ such that $w_i > 0$ and $w_j < 0$. Recall from Definition S4.2 that $[\mathfrak{s}(cw)]_{(i,j)} = \text{sgn}\,(1 - c(w_i - w_j))$. Thus, as $c$ ranges over $\mathbb{R}$, we have that $[\mathfrak{s}(cw)]_{(i,j)}$ takes on all three values in $\{-1, 0, 1\}$. However, by Lemma S4.4 item 2, $[\mathfrak{s}(cw)]_{(i,j)}$ can only take on at most two distinct values in $\{-1, 0, 1\}$. $\qquad \square$

Before proceeding, we recall a definition:

**Definition S4.8.** A *polyhedron* $P$ in $\mathbb{R}^k$ is a set of the form $P = \{x \in \mathbb{R}^k : \langle a^{(n)}, x \rangle \leq b^{(n)}, \forall n \in [m]\}$ where $m$ is a positive integer, $a^{(n)} \in \mathbb{R}^k$ and $b^{(n)} \in \mathbb{R}$ for all $n \in [m]$. For each $n \in [m]$, the tuple $(a^{(n)}, b^{(n)})$ is called a *constraint* of $P$. A point $x \in P$ is a *basic feasible solution* (BFS) if there exists $n_1, \ldots, n_k \in [m]$ such that

1. $\langle a^{(n_i)}, x \rangle = b^{(n_i)}$ for all $i \in [k]$, and

2. $\mathcal{A} := \{a^{(n_1)}, \ldots, a^{(n_k)}\}$ is a basis for $\mathbb{R}^k$.

By Bertsimas and Tsitsiklis [1, Theorem 2.6] and [1, Theorem 2.3], a polyhedron which does not contain any line always have a BFS. Earlier, we proved that $H_0 \cap P_\theta$ does not contain any line. Hence, $H_0 \cap P_\theta$ contains a BFS. For the remainder of this proof, let $x \in \mathbb{R}^k$ be such a BFS with associated basis $\mathcal{A} = \{a^{(n_1)}, \ldots, a^{(n_k)}\}$ as in Definition S4.8.

Let $e^i \in \mathbb{R}^k$ be the $i$-th elementary basis vector in $\mathbb{R}^k$. By definition of $P_\theta \cap H_0$, we have

$$\mathcal{A} \subseteq \{e^i - e^j : (i, j) \in I\} \cup \{\mathbf{1}^k\}$$

where we recall that $I$ is as in Definition S4.5. Observe that $\langle \mathbf{1}^k, e^i - e^j \rangle = 0$ for all $(i, j) \in I$. Hence, we must have that $\mathbf{1}^k \in \mathcal{A}$, since otherwise $\mathcal{A}$ cannot span $\mathbb{R}^k$. This implies that we necessarily have $\mathbf{1}^k \in \mathcal{A}$. Without the loss of generality, let $a^{(n_k)} = \mathbf{1}^k$. Since $\mathcal{A}$ is linearly independent, we have

$$\mathcal{B} := \mathcal{A} \setminus \{a^{(n_k)}\} = \{a^{(n_1)}, \ldots, a^{(n_{k-1})}\} \subseteq \{e^i - e^j : (i, j) \in I\}.$$

Now, for each $i \in [k-1]$, let $(t_i, h_i) \in I$ be such that $a^{(n_i)} = e^{t_i} - e^{h_i}$. By the definition of $P_\theta$, we have $\langle a^{(n_i)}, x \rangle = x_{t_i} - x_{h_i} = \pm 1$. Note that this implies that $x$ is not a scalar multiple of $\mathbf{1}^k$.

Next, consider the directed graph $G$ with vertices $V(G) = [k]$ and edges are $E(G) = \{(t_i, h_i) : i \in [k-1]\}$. Since $\mathcal{B}$ is linearly independent, we observe that if $(t_i, h_i) \in E(G)$, then $(h_i, t_i) \notin E(G)$. Let $G^u$ be the undirected graph obtained from $G$ by forgetting the edge orientations. By the preceding observation, we have $|E(G^u)| = k - 1$. An undirected edge is denoted as $\{\alpha, \beta\} \in E(G^u)$.

Observe that if $\{\alpha, \beta\} \in E(G^u)$, then $x_\alpha - x_\beta = \pm 1$.

**Lemma S4.9.** $G^u$ *is a tree, i.e., a connected graph without cycles.*

*Proof.* Note that $G^u$ does not contain any cycles. To see this, note that if $G^u$ had a cycle, then $\mathcal{A}$ cannot be linearly independent. Thus, $G^u$ is a disjoint union of trees $\{T_1, \ldots, T_f\}$ where $f$ is a positive integer. Since each $T_i$ is a tree, we have $|E(T_i)| = |V(T_i)| - 1$. On the other hand, we have

$$
\begin{aligned}
k - 1 &= |E(G^u)| \\
&= |E(T_1)| + \cdots + |E(T_f)| \\
&= |V(T_i)| + \cdots + |V(T_f)| - f \\
&= |V(G^u)| - f \\
&= k - f
\end{aligned}
$$

which implies that $f = 1$. In other words, $G^u$ is a tree to begin with. $\qquad\square$

Although we know that $G^u$ is a tree, we only need the fact that $G^u$ is connected.

Let $\alpha, \beta \in V(G^u)$. A *path* of length $l$ from $\alpha$ to $\beta$ is a sequence $\phi_1, \ldots, \phi_l \in V(G^u)$ such that

1. $\phi_1 = \alpha$ and $\phi_l = \beta$

2. $\{\phi_i, \phi_{i+1}\} \in E(G^u)$ for all $i \in [m-1]$.

The fact that $G^u$ is connected implies that there exists a path between any two vertices $\alpha, \beta \in V(G^u)$. Define $\overline{x} := \max x$ and $\underline{x} := \min x$.

**Lemma S4.10.** *For all $\beta \in [k]$, we have $\overline{x} - x_\beta \in \mathbb{Z}$.*

*Proof.* Let $\alpha \in \arg\max x$ and consider a path $\phi_1, \ldots, \phi_l \in V(G^u)$ from $\alpha$ to $\beta$. Observe that $x_\alpha - x_\beta = \sum_{i \in [l-1]} x_{\phi_i} - x_{\phi_{i+1}}$. Since $\{\phi_i, \phi_{i+1}\} \in E(G^u)$, we have $x_{\phi_i} - x_{\phi_{i+1}} = \pm 1$. This proves that $x_\alpha - x_\beta \in \mathbb{Z}$. $\qquad\square$

Let $D := \overline{x} - \underline{x}$. Since $x_\beta \geq \underline{x}$, we have $0 \leq \overline{x} - x_\beta \leq D$. Apply Lemma S4.10 with $\beta \in \arg\min x$, we get $\overline{x} - \underline{x} = D \in \mathbb{Z}$. In summarize, we have proven that

$$\{x_\beta - \overline{x} : \beta \in [k]\} \subseteq \{-D, -D+1, \ldots, -1, 0\}. \tag{S43}$$

Below, we will show that the inclusion in eq. (S43) is in fact an equality.

Next, let $\overline{\varrho} \in \arg\max x$ and $\underline{\varrho} \in \arg\min x$. Let $\phi_1, \ldots, \phi_l \in V(G^u)$ be a path between $\overline{\varrho}$ and $\underline{\varrho}$. Note that by definition we have

1. $x_{\phi_1} = \overline{x}$ and $x_{\phi_l} = \underline{x}$,

2. $x_{\phi_i} - x_{\phi_{i+1}} = \pm 1$ for all $i \in [l-1]$.

Consider the sequence of numbers

$$S := (\underbrace{x_{\phi_1} - \overline{x}}_{=-D}, x_{\phi_2} - \overline{x}, \ldots, x_{\phi_{l-1}} - \overline{x}, \underbrace{x_{\phi_l} - \overline{x}}_{=0}).$$

Notice that the difference between consecutive entries of $S$ is $\pm 1$. Thus, the sequence $S$ takes on every value in $\{-D, -D+1, \ldots, -1, 0\}$ at least once. This proves that eq. (S43) holds with equality, i.e.,

$$\{x_\beta - \overline{x} : \beta \in [k]\} = \{-D, -D+1, \ldots, -1, 0\}. \tag{S44}$$

Now, let $\sigma \in \mathfrak{S}_k$ be the element such that $\sigma x$ is monotonic non-increasing. Earlier, we argued that $x$ is not a scalar multiple of $\mathbf{1}^k$. Thus, eq. (S44) implies that $\sigma x - \overline{x}\mathbf{1}^k \in \mathcal{C}_{\mathbb{Z}}$. Consequently, we have $x - \overline{x}\mathbf{1}^k \in \mathfrak{S}_k \mathcal{C}_{\mathbb{Z}}$. Since $x \in P_\theta$, we have $x - \overline{x}\mathbf{1}^k \in P_\theta$ as well. This proves that $P_\theta \cap \mathfrak{S}_k \mathcal{C}_{\mathbb{Z}}$ is nonempty, which concludes the proof of Lemma S4.4. $\qquad\square$

## S4.2  Proof of Proposition 4.3

*Proof of Proposition 4.3.* Let $m = |\mathcal{OP}_k|$. Index the elements of $\mathcal{OP}_k$ by $[m]$, i.e.,

$$\mathcal{OP}_k = \{\mathbf{S}^1, \ldots, \mathbf{S}^m\}.$$

For each $i \in [m]$, let $p^{(i)} \in \Delta^k$ be such that $\{\mathbf{S}^i\} = \arg\min_{\mathbf{S} \in \mathcal{OP}_k} \langle p, \ell(\mathbf{S}) \rangle$. The existence of such $p^{(i)}$s was confirmed by computer search for $k \in \{3, \ldots, 15\}$. Equivalently, $\mathbf{S}^i$ is the unique element of $\mathcal{OP}_k$ such that

$$\langle p^{(i)}, \ell(\mathbf{S}^i) \rangle = \underline{\ell}(p^{(i)}) = \underline{L}(p^{(i)}) \tag{S45}$$

where the second equality is by Corollary 3.9.

Next, suppose $L$ embeds another discrete loss $\lambda : \mathcal{R} \to \mathbb{R}^k_+$ with embedding map $\chi : \mathcal{R} \to \mathbb{R}^k$. Our goal is to show that $|\mathcal{R}| \geq |\mathcal{OP}_k|$. To this end, let $\mathcal{R} = \{r^1, \ldots, r^n\}$. Since $L$ embeds $\lambda$ via $\chi$, we have by definition that $\underline{L}(p) = \underline{\lambda}(p) = \min_{r \in \mathcal{R}} \langle p, L(\chi(r)) \rangle$. In particular, for a fixed $i \in [m]$, there exists $\iota(i) \in [n]$ such that $\underline{L}(p^{(i)}) = \langle p^{(i)}, L(\chi(r^{\iota(i)})) \rangle$. Note that this defines a mapping

$$\iota : [m] \to [n]. \tag{S46}$$

Let $v^{(i)} := \chi(r^{\iota(i)})$. Combined with eq. (S45), we have

$$\langle p^{(i)}, L(v^{(i)}) \rangle = \underline{L}(p^{(i)}) = \underline{\ell}(p^{(i)}). \tag{S47}$$

Consider $\{P_\theta\}_{\theta \in \Theta}$ as in Lemma S4.6. For each $v \in \mathbb{R}^k$, let $\theta(v) \in \Theta$ be the unique element such that $v \in \operatorname{relint}\left(P_{\theta(v)}\right)$. The existence and uniqueness of $\theta(v)$ is guaranteed by Lemma S4.4 item 4.

By eq. (S47), we have $v^{(i)} \in \arg\min_{v \in \mathbb{R}^k} \langle p^{(i)}, L(v) \rangle$. By Lemma S4.6, the function $v \mapsto \langle p^{(i)}, L(v) \rangle$ is affine over the domain $P_{\theta(v^{(i)})}$. Furthermore, it is minimized at $v^{(i)} \in \operatorname{relint}(P_{\theta(v)})$. Thus, by Ben-Tal and Nemirovski [3, Lemma 1.2.2], the function $v \mapsto \langle p^{(i)}, L(v) \rangle$ is constant over the domain $v \in P_{\theta(v^{(i)})}$. Since $v^{(i)} \in P_{\theta(v^{(i)})}$ and $\langle p^{(i)}, L(v^{(i)}) \rangle = \underline{L}(p^{(i)})$ by eq. (S47), we have

$$\langle p^{(i)}, L(v) \rangle = \underline{L}(p^{(i)}), \ \forall v \in P_{\theta(v^{(i)})} \tag{S48}$$

Next, recall that $P_\theta \cap \mathfrak{S}_k \mathcal{C}_\mathbb{Z}$ is nonempty for all $\theta \in \Theta$. In particular, $P_{\theta(v^{(i)})} \cap \mathfrak{S}_k \mathcal{C}_\mathbb{Z}$ is nonempty. By Proposition 3.6, we have $\mathfrak{S}_k \mathcal{C}_\mathbb{Z} = \varphi(\mathcal{OP}_k)$. All elements of $P_{\theta(v^{(i)})} \cap \mathfrak{S}_k \mathcal{C}_\mathbb{Z}$ are of the form $\varphi(\mathbf{S})$ for some $\mathbf{S} \in \mathcal{OP}_k$. Fix such an $\mathbf{S}$ so that $\varphi(\mathbf{S}) \in P_{\theta(v^{(i)})} \cap \mathfrak{S}_k \mathcal{C}_\mathbb{Z}$. Now,

$$\langle p^{(i)}, L(\varphi(\mathbf{S})) \rangle \overset{\text{eq. (S48)}}{=} \underline{L}(p^{(i)}) \overset{\text{eq. (S47)}}{=} \underline{\ell}(p^{(i)}).$$

Recall from right before eq. (S45), we have that $\mathbf{S}^i$ is the unique element of $\mathcal{OP}_k$ such that $\langle p^{(i)}, L(\varphi(\mathbf{S}^{(i)})) \rangle = \underline{\ell}(p^{(i)})$. This proves that $\mathbf{S} = \mathbf{S}^i$. Thus, we have shown that

$$P_{\theta(v^{(i)})} \cap \mathfrak{S}_k \mathcal{C}_\mathbb{Z} = \{\varphi(\mathbf{S}^i)\}. \tag{S49}$$

Finally, we are now ready to prove that $n = |\mathcal{R}| \geq |\mathcal{OP}_k| = m$. It suffices to show that the mapping $\iota : [m] \to [n]$ defined at eq. (S46) is injective. Suppose that there exists distinct $i, j \in [m]$ such that $\iota(i) = \iota(j)$. Then

$$r^{\iota(i)} = r^{\iota(j)}$$
$$\implies v^{(i)} = v^{(j)} \quad \because \text{definition of } v^{(i)} := \chi(r^{\iota(i)})$$
$$\implies \theta(v^{(i)}) = \theta(v^{(j)})$$
$$\implies P_{\theta(v^{(i)})} \cap \mathfrak{S}_k \mathcal{C}_\mathbb{Z} = P_{\theta(v^{(j)})} \cap \mathfrak{S}_k \mathcal{C}_\mathbb{Z}$$
$$\implies \{\varphi(\mathbf{S}^i)\} = \{\varphi(\mathbf{S}^j)\} \quad \because \text{eq. (S49)}$$
$$\implies \varphi(\mathbf{S}^i) = \varphi(\mathbf{S}^j)$$
$$\implies \mathbf{S}^i = \mathbf{S}^j \quad \because \varphi \text{ is a bijection}$$

which contradicts $i \neq j$. Thus, we have that $\iota : [m] \to [n]$ is injective which implies that $n \geq m$. $\quad\square$

## S5 The argmax link

**Definition S5.1.** For $\sigma \in \mathfrak{S}_k$ and $\mathbf{S} = (S_1, \ldots, S_l) \in \mathcal{OP}_k$, define $\sigma(\mathbf{S}) \in \mathcal{OP}_k$ by

$$\sigma(\mathbf{S}) = (\sigma(S_1), \ldots, \sigma(S_l))$$

where $\sigma(S_i) = \{\sigma(j) : j \in S_i\}$ for each $i \in [l]$.

**Lemma S5.2.** *For $\sigma \in \mathfrak{S}_k$ and $\mathbf{S} = (S_1, \ldots, S_l) \in \mathcal{OP}_k$, we have*

$$\sigma'\varphi(\mathbf{S}) = \varphi(\sigma(\mathbf{S})).$$

*Proof.* By definition, we have

$$[\varphi(\sigma(\mathbf{S}))]_j = -(i-1), \ \forall j \in \sigma(S_i).$$

Since $j \in \sigma(S_i) \iff \sigma^{-1}(j) \in S_i$, we have

$$[\varphi(\sigma(\mathbf{S}))]_j = -(i-1), \ \forall j \in [k] : \sigma^{-1}(j) \in S_i.$$

Introduce the change of variable $m = \sigma^{-1}(j)$, we have

$$[\varphi(\sigma(\mathbf{S}))]_{\sigma(m)} = -(i-1), \ \forall m \in S_i.$$

On the other hand, we have

$$[\sigma'\varphi(\mathbf{S})]_{\sigma(m)} = [\varphi(S)]_{\sigma'\sigma(m)} = [\varphi(S)]_m = -(i-1), \ \forall m \in S_i.$$

This proves that $\sigma'\varphi(\mathbf{S}) = \varphi(\sigma\mathbf{S})$. $\qquad\square$

**Corollary S5.3.** *For all $\mathbf{S} \in \mathcal{OP}_k$ and $\sigma \in \mathfrak{S}_k$, we have $\sigma\ell(\mathbf{S}) = \ell(\sigma'\mathbf{S})$.*

*Proof.* Since $\Delta^k$ spans $\mathbb{R}^k$, it suffices to check that $\langle p, \sigma\ell(\mathbf{S}) \rangle = \langle p, \ell(\sigma'\mathbf{S}) \rangle$ for all $p \in \Delta^k$. To this end, we have

$$
\begin{aligned}
\langle p, \ell(\sigma'\mathbf{S}) \rangle &= \langle p, L(\varphi(\sigma'\mathbf{S})) \rangle && \because \text{Theorem 3.8} \\
&= \langle p, L(\sigma\varphi(\mathbf{S})) \rangle && \because \text{Lemma S5.2} \\
&= \langle p, \sigma L(\varphi(\mathbf{S})) \rangle && \because \text{Corollary S3.5} \\
&= \langle \sigma'p, L(\varphi(\mathbf{S})) \rangle \\
&= \langle \sigma'p, \ell(\mathbf{S}) \rangle && \because \text{Theorem 3.8} \\
&= \langle p, \sigma\ell(\mathbf{S}) \rangle
\end{aligned}
$$

as desired. $\qquad\square$

For $p \in \Delta^k$, define

$$\gamma(p) := \arg\min_{\mathbf{S} \in \mathcal{OP}_k} \langle p, \ell(\mathbf{S}) \rangle, \tag{S50}$$

$$\Gamma(p) := \arg\min_{v \in \mathbb{R}^k} \langle p, L(v) \rangle. \tag{S51}$$

**Lemma S5.4.** *Let $p \in \Delta^k_\downarrow$, $v \in \Gamma(p)$, and $\sigma$ be such that $\sigma v \in \mathbb{R}^k_\downarrow$. Then $\sigma p = p$ and $\sigma v \in \Gamma(p)$.*

*Proof.* Let $i \in [k-1]$ be such that $v_i < v_{i+1}$. We first prove that $p_i = p_{i+1}$. Let $\tau = \sigma_{(i,i+1)}$. Since $\tau$ is a transposition, we have $\tau' = \tau$. Now,

$$
\begin{aligned}
0 &\leq \langle p, L(\tau v) \rangle - \langle p, L(v) \rangle && \because \text{Optimality of } v \\
&= \langle p, \tau L(v) \rangle - \langle p, L(v) \rangle && \because \text{Corollary S3.5} \\
&= \langle \tau p, L(v) \rangle - \langle p, L(v) \rangle && \because \tau' = \tau. \\
&= (p_{i+1} - p_i)[L(v)]_i + (p_i - p_{i+1})[L(v)]_{i+1} \\
&= (p_{i+1} - p_i)([L(v)]_i - [L(v)]_{i+1})
\end{aligned}
$$

By Lemma S3.6, we have $[L(v)]_i - [L(v)]_{i+1} > 0$. By assumption, we have $p_i \geq p_{i+1}$. If we have $p_i > p_{i+1}$, then

$$\underbrace{(p_{i+1} - p_i)}_{<0}\underbrace{([L(v)]_i - [L(v)]_{i+1})}_{>0} < 0$$

which is a contradiction. Hence, we must have $p_i = p_{i+1}$. Repeating the proof with the update $v \leftarrow \tau v$, we obtain a composition of transpositions

$$\sigma := \sigma_{(i_1, i_1+1)}\sigma_{(i_2, i_2+1)} \cdots \sigma_{(i_m, i_m+1)}$$

such that $\sigma v \in \mathbb{R}_{\downarrow}^k$ and $\sigma p = p$. Finally,

$$\underline{L}(p) = \langle p, L(v) \rangle = \langle p, \sigma'\sigma L(v) \rangle = \langle \sigma p, L(\sigma v) \rangle = \langle p, L(\sigma v) \rangle$$

implies that $\sigma v \in \Gamma(p)$. $\qquad\square$

**Lemma S5.5.** *Let* $\sigma \in \mathfrak{S}_k$ *and* $v \in \mathbb{R}^k$. *Then* $\arg\max \sigma v = \sigma^{-1}(\arg\max v)$.

*Proof.* Let $M = \max v = \max \sigma v$.

$$\begin{aligned}
\arg\max \sigma v &= \{j \in [k] : [\sigma v]_j = M\} \\
&= \{j \in [k] : [v]_{\sigma(j)} = M\}.
\end{aligned}$$

On the other hand,

$$\begin{aligned}
\sigma^{-1}(\arg\max v) &= \{j \in [k] : \sigma(j) \in \arg\max v\} \\
&= \{j \in [k] : [v]_{\sigma(j)} = M\} \\
&= \arg\max \sigma v
\end{aligned}$$

as desired. $\qquad\square$

**Lemma S5.6.** *Let* $p \in \Delta_{\downarrow}^k$ *be such that* $\max p > \frac{1}{k}$. *Let* $v \in \Gamma(p)$, *then there exists* $\mathbf{S} = (S_1, \ldots, S_l) \in \gamma(p)$ *such that* $\arg\max v \subseteq S_1$.

*Proof.* Recall by definition, $v \in \Gamma(p)$ if and only if $\underline{L}(p) = \langle p, L(v) \rangle$. We first claim that $v$ is not a scalar multiple of the all ones vector. Suppose it is, then $\underline{L}(p) = \langle p, L(v) \rangle = \langle p, L(0) \rangle$ by Lemma S3.1, which implies that $0 \in \Gamma(p)$. Now, by Lemma S3.20, we have $0 \notin \Gamma(p)$ since $\min p < \frac{1}{k}$ by the assumption that $\max p > \frac{1}{k}$. This is a contradiction. Hence, the claim is proved.

Next, let $n = |\arg\max v|$. By our claim that $v$ is non-constant, we have that $n \in [k-1]$. Let $\sigma \in \mathfrak{S}_k$ be such that $\sigma v \in \mathbb{R}_{\downarrow}^k$. Thus, by construction, we have $\arg\max v = [n]$. Hence, we have, by Lemma S5.5,

$$[n] = \arg\max \sigma v = \sigma^{-1}(\arg\max v)$$

or, equivalently, $\arg\max v = \sigma([n])$. Since $n = |\arg\max v| \in [k-1]$, $v$ is feasible for the right hand side of eq. (S6). Thus, we have

$$\underline{L}(p) = \underline{L}^n(p).$$

By Lemma S3.19

$$\underline{L}^n(p) = \min_{w \in \mathcal{C}_{\mathbb{Z}} : w_n = 0} \langle p, L(w) \rangle. \tag{S52}$$

Let $w^*$ be a minimizer of the above optimization. Since $w^* \in \mathcal{C}_{\mathbb{Z}}$, consider $\mathbf{S} = (S_1, \ldots, S_l) := \tilde{\psi}(w^*)$. Hence, by the definition of $\tilde{\psi}$, we have that $S_1 = \arg\max w^*$. Note that

$$\begin{aligned}
\underline{L}(p) = \underline{L}^n(p) &= \langle p, L(w^*) \rangle \\
&= \langle p, L(\varphi(\mathbf{S})) \rangle \quad \because \text{Proposition 3.6} \\
&= \langle p, \ell(\mathbf{S}) \rangle \quad \because \text{Theorem 3.8} \\
&= \langle \sigma p, \ell(\mathbf{S}) \rangle \quad \because \sigma p = p \text{ by Lemma S5.4} \\
&= \langle p, \sigma'\ell(\mathbf{S}) \rangle \\
&= \langle p, \ell(\sigma\mathbf{S}) \rangle \quad \because \text{Corollary S5.3.}
\end{aligned}$$

Putting it all together, we have

$$\langle p, \ell(\sigma\mathbf{S}) \rangle = \underline{L}(p) = \underline{\ell}(p)$$

where the second equality follows from Corollary 3.9. This proves that $\sigma \mathbf{S} \in \gamma(p)$. Note that since $w^*$ is feasible for the optimization on the right hand side of eq. (S52), we have $\arg\max w^* = \{i \in [k] : w_i^* = 0\} \supseteq [n]$. Furthermore, recall that $S_1 = \arg\max w^*$. Putting it all together, we have $\sigma(S_1) \supseteq \sigma([n]) = \arg\max v$. Thus, $\sigma(\mathbf{S})$ satisfies the desired conditions. $\quad\square$

**Lemma S5.7.** *For all $p \in \Delta^k$ and $\sigma \in \mathfrak{S}_k$, we have*

$$\mathbf{S} \in \gamma(\sigma p) \iff \sigma \mathbf{S} \in \gamma(p), \tag{S53}$$

$$v \in \Gamma(\sigma p) \iff \sigma' v \in \Gamma(p). \tag{S54}$$

*Proof.* We first prove eq. (S53). Let $\mathbf{S} \in \gamma(\sigma p)$. Then

$$\begin{aligned}
\underline{\ell}(\sigma p) &= \langle \sigma p, \ell(\mathbf{S}) \rangle \\
&= \langle p, \sigma' \ell(\mathbf{S}) \rangle \\
&= \langle p, \ell(\sigma \mathbf{S}) \rangle \quad \because \text{Corollary S5.3} \\
&\geq \underline{\ell}(p).
\end{aligned}$$

By the same argument, we have $\underline{\ell}(p) \geq \underline{\ell}(\sigma p)$. Thus, $\underline{\ell}(p) = \underline{\ell}(\sigma p)$ and $\sigma \mathbf{S} \in \gamma(p)$. This proves the $\implies$ direction eq. (S53). To prove the other direction, we first write $p = \sigma' \sigma p$ and note that

$$\sigma \mathbf{S} \in \gamma(\sigma' \sigma p) \implies \sigma' \sigma \mathbf{S} \in \gamma(\sigma p) \iff \mathbf{S} \in \gamma(\sigma p).$$

Next, we prove eq. (S54). By Lemma S3.7, we have $\underline{L}(\sigma p) = \underline{L}(p)$. Let $v \in \Gamma(\sigma p)$, then

$$\begin{aligned}
\underline{L}(p) = \underline{L}(\sigma p) &= \langle \sigma p, L(v) \rangle \\
&= \langle p, \sigma' L(v) \rangle \\
&= \langle p, L(\sigma' v) \rangle \quad \because \text{Corollary S3.5.}
\end{aligned}$$

Thus, $\sigma' v \in \Gamma(p)$. This proves the $\implies$ direction of eq. (S54). For the other direction,

$$\sigma' v \in \Gamma(\sigma' \sigma p) \implies \sigma \sigma' v \in \Gamma(\sigma p) \iff v \in \Gamma(\sigma p).$$

$\quad\square$

## S5.1 Proof of Theorem 5.2

*Proof of Theorem 5.2.* Let $\sigma \in \mathfrak{S}_k$ be such that $\sigma p \in \Delta_\downarrow^k$. By Lemma S5.7, we have $\sigma v \in \Gamma(\sigma p)$. Then by Lemma S5.6, there exists $\mathbf{S} = (S_1, \ldots, S_l) \in \gamma(\sigma p)$ such that $S_1 \supseteq \arg\max \sigma v = \sigma^{-1}(\arg\max v)$, where the equality is due to Lemma S5.5. Applying $\sigma$, to both side, we have $\sigma S_1 \supseteq \arg\max v$. By Lemma S5.7, we have $\sigma \mathbf{S} \in \gamma(p)$. Hence, we are done. $\quad\square$

**Lemma S5.8.** *Let $p \in \Delta_\downarrow^k$ be such that $\arg\max p = \{1\}$ and $\mathbf{S} = (S_1, \ldots, S_l) \in \gamma(p)$. Then $1 \in S_1$.*

*Proof.* Let $v = \varphi(\mathbf{S})$. Since $\mathbf{S}$ is nontrivial, we have $\max v > \min v$. By construction, we have $\arg\max v = S_1$. Hence, if $1 \notin S_1$, then there exists some $j \in \{2, \ldots, k\}$ such that $v_j > v_1$. Then Lemma S3.6 implies that $[L(v)]_1 > [L(v)]_j$ and so

$$\langle p, L(v) \rangle - \langle p, \sigma_j L(v) \rangle = (p_1 - p_j)([L(v)]_1 - [L(v)]_j) > 0.$$

But $\underline{\ell}(p) = \langle p, \ell(\mathbf{S}) \rangle = \langle p, L(v) \rangle$ and

$$\langle p, \sigma_j L(v) \rangle = \langle p, L(\sigma_j v) \rangle = \langle p, L(\sigma_j \varphi(\mathbf{S})) \rangle = \langle p, L(\varphi(\sigma_j \mathbf{S})) \rangle = \langle p, \ell(\sigma_j \mathbf{S}) \rangle$$

Thus, we have

$$\langle p, \ell(\mathbf{S}) \rangle - \langle p, \ell(\sigma_j \mathbf{S}) \rangle > 0$$

which contradicts that $\mathbf{S} \in \gamma(p)$. $\quad\square$

**Definition S5.9.** A $\mathfrak{S}_k$-*invariant property* is a boolean function

$$\mathcal{B} : \Delta^k \to \{\texttt{true}, \texttt{false}\} \tag{S55}$$

such that $\mathcal{B}(p) \implies \mathcal{B}(\sigma p)$ for all $\sigma \in \mathfrak{S}_k$ and $p \in \Delta^k$. Here, "$\implies$" denotes logical implication.

**Lemma S5.10.** *Let $\mathcal{B}$ and $\mathcal{C}$ be $\mathfrak{S}_k$-invariant properties. Suppose that for all $p \in \Delta_\downarrow^k$, $\mathcal{B}(p)$ implies $\mathcal{C}(p)$. Then for all $p \in \Delta^k$, we have $\mathcal{B}(p)$ implies $\mathcal{C}(p)$.*

*Proof.* Let $p \in \Delta^k$ be arbitrary. Pick $\sigma$ such that $\sigma p \in \Delta_\downarrow^k$. Then

$$\mathcal{B}(p) \implies \mathcal{B}(\sigma p) \implies \mathcal{C}(\sigma p) \implies \mathcal{C}(p)$$

where for the first and last implications we used the $\mathfrak{S}_k$-invariance property of $\mathcal{B}$ and $\mathcal{C}$, and for the implication in the middle we used the assumption in the lemma. $\qquad\square$

**Lemma S5.11.** *Let $p \in \Delta^k$. Consider the statement $\mathcal{B}_1(p)$ which returns* true *if and only if*

$$\text{for all } \mathbf{S} \in \gamma(p), \; |S_1| = 1 \text{ and } S_1 = \arg\max p. \tag{S56}$$

*Then $\mathcal{B}_1$ is a $\mathfrak{S}_k$-invariant property.*

*Proof.* Let $p \in \Delta^k$ and $\sigma \in \mathfrak{S}_k$. Suppose $\mathcal{B}_1(p)$ is true. We need to show that $\mathcal{B}_1(\sigma' p)$ is true. Let $\mathbf{S} \in \gamma(\sigma p)$. By Lemma S5.7, we have $\sigma \mathbf{S} \in \gamma(p)$. Since $\mathcal{B}_1(p)$ is true, we have $|\sigma(S_1)| = 1$ and $\sigma(S_1) = \arg\max p$. Thus, we immediately get that $|S_1| = 1$. By Lemma S5.5, we have $S_1 = \sigma^{-1}(\arg\max p) = \arg\max \sigma p$. The two preceding facts is equivalent to $\mathcal{B}_1(p)$ being true, by definition. $\qquad\square$

## S5.2 Proof of Proposition 5.3

*Proof of Proposition 5.3.* By Lemma S5.11 and Lemma S5.10, we may assume $p \in \Delta_\downarrow^k$. Lemma S5.8 implies that $1 \in S_1$. If $|S_1| = 1$, then $S_1 = \{1\}$ and the result is proven. Below, suppose $|S_1| > 1$. We define

$$S_1' = \{1\}, \quad S_1'' = S_1 \setminus \{1\}.$$

Define

$$\mathbf{S}' = (S_1', S_1'', S_2, \dots, S_l) \in \mathcal{OP}_k.$$

We claim that $\langle p, \ell(\mathbf{S}') \rangle < \langle p, \ell(\mathbf{S}) \rangle$. Given the claim, we would have a contradiction that $\mathbf{S} \in \gamma(p)$ and so $|S_1| = 1$ must be true. Let $Y \sim p$ and define

$$\beta := \sum_{j=1}^{l-1} |S_1 \cup \dots \cup S_{j+1}| \Pr(Y \notin S_1 \cup \dots \cup S_j)$$

Observe that

$$\begin{aligned}
\langle p, \ell(\mathbf{S}') \rangle &= |S_1'| - 1 + |S_1' \cup S_1''| \Pr(Y \notin S_1') + \beta \\
&= |S_1| \Pr(Y \neq 1) + \beta \\
&< \frac{1}{2}|S_1| + \beta.
\end{aligned}$$

On the other hand, we have

$$\langle p, \ell(\mathbf{S}) \rangle = |S_1| - 1 + \beta.$$

Hence, we have

$$\begin{aligned}
\langle p, \ell(\mathbf{S}) \rangle - \langle p, \ell(\mathbf{S}') \rangle &= |S_1| - 1 - |S_1| \Pr(Y \neq 1) \\
&> |S_1| - 1 - \frac{1}{2}|S_1| \\
&= \frac{1}{2}|S_1| - 1 \\
&\geq \frac{2}{2} - 1 \\
&= 0.
\end{aligned}$$

which proves the claim. $\qquad\square$

## S5.3 Proof of Proposition 5.4

*Proof of Proposition 5.4.* Since $\arg\max p = \{j^*\}$, we have $(\{j^*\}, [k] \setminus \{j^*\}) = (\arg\max p, [k] \setminus \arg\max p)$. We check that the statement below defines a $\mathfrak{S}_k$-invariant property:

$$\text{``}p \text{ satisfies } (\arg\max p, [k] \setminus \arg\max p) \text{ is the unique element of } \gamma(p)\text{.''} \qquad \text{(S57)}$$

Let $p$ satisfy eq. (S57). By Lemma S5.7, we have $\sigma^{-1}(\arg\max p, [k] \setminus \arg\max p)$ is the unique element of $\gamma(\sigma p)$. By definition,

$$\sigma^{-1}(\arg\max p, [k] \setminus \arg\max p) = (\sigma^{-1} \arg\max p, \sigma^{-1}([k] \setminus \arg\max p)).$$

By Lemma S5.5, we have $\sigma^{-1} \arg\max p = \arg\max \sigma p$. Thus, we have

$$\sigma^{-1}(\arg\max p, [k] \setminus \arg\max p) = (\arg\max \sigma p, [k] \setminus \arg\max \sigma^{-1} p)$$

is the unique element of $\gamma(\sigma p)$. In other words, $\sigma p$ satisfies eq. (S57), as desired.

Furthermore, "$p$ satisfies the symmetric noise condition." is obviously $\mathfrak{S}_k$-invariant. Hence, by Lemma S5.11 and Lemma S5.10, we may assume $p \in \Delta_\downarrow^k$. Pick $\mathbf{S} = (S_1, \ldots, S_l) \in \gamma(p)$. Lemma S5.8 implies that $1 \in S_1$. By Definition 2.1 of $\mathcal{OP}_k$, we have $l \geq 2$. We first show that $l = 2$ by contradiction. Suppose that $l > 2$. Define $\mathbf{S}' = (S_1', \ldots, S_{l-1}')$ where

$$S_1' := S_1, \quad S_2' := S_2 \cup S_3, \quad S_j' := S_{j+1}, \forall j \in \{3, \cdots, l-1\}.$$

Let $Y \sim p$ and

$$\beta := \sum_{j=3}^{l-1} |S_1 \cup \cdots \cup S_{j+1}| \Pr(Y \notin S_1 \cup \cdots \cup S_j).$$

Then we have

$$\langle p, \ell(\mathbf{S}) \rangle = |S_1| - 1 + |S_1 \cup S_2| \Pr(Y \notin S_1)$$
$$+ |S_1 \cup S_2 \cup S_3| \Pr(Y \notin S_1 \cup S_2) + \beta$$

and

$$\langle p, \ell(\mathbf{S}') \rangle = |S_1'| - 1 + |S_1' \cup S_2'| \Pr(Y \notin S_1')$$
$$+ \sum_{j=2}^{l-2} |S_1' \cup \cdots \cup S_{j+1}'| \Pr(Y \notin S_1' \cup \cdots \cup S_j')$$
$$= |S_1| - 1 + |S_1 \cup S_2 \cup S_3| \Pr(Y \notin S_1)$$
$$+ \sum_{j=2}^{l-2} |S_1 \cup \cdots \cup S_{j+2}| \Pr(Y \notin S_1 \cup \cdots \cup S_{j+1})$$
$$= |S_1| - 1 + |S_1 \cup S_2 \cup S_3| \Pr(Y \notin S_1)$$
$$+ \sum_{j=3}^{l-1} |S_1 \cup \cdots \cup S_{j+1}| \Pr(Y \notin S_1 \cup \cdots \cup S_j)$$
$$= |S_1| - 1 + |S_1 \cup S_2 \cup S_3| \Pr(Y \notin S_1) + \beta$$

Putting it all together, we have

$$\langle p, \ell(\mathbf{S}) \rangle - \langle p, \ell(\mathbf{S}') \rangle = |S_1 \cup S_2| \Pr(Y \notin S_1)$$
$$+ |S_1 \cup S_2 \cup S_3| \Pr(Y \notin S_1 \cup S_2)$$
$$- |S_1 \cup S_2 \cup S_3| \Pr(Y \notin S_1)$$
$$= |S_1 \cup S_2 \cup S_3| \Pr(Y \notin S_1 \cup S_2)$$
$$- |S_3| \Pr(Y \notin S_1).$$

Define $s_i := |S_i|$ for each $i \in [l]$. Then

$$|S_1 \cup S_2 \cup S_3| \Pr(Y \notin S_1 \cup S_2) = (s_1 + s_2 + s_3)(k - s_1 - s_2)\frac{1 - \alpha}{k - 1}$$

and
$$|S_3| \Pr(Y \notin S_1) = s_3(k - s_1)\frac{1 - \alpha}{k - 1}.$$

Now, we have
$$
\begin{aligned}
& (s_1 + s_2 + s_3)(k - s_1 - s_2) - s_3(k - s_1) \\
&= ((s_1 + s_2) + s_3)((k - s_1) - s_2) - s_3(k - s_1) \\
&= (s_1 + s_2)(k - s_1) - s_2(s_1 + s_2) - s_2 s_3 \\
&= (s_1 + s_2)k - (s_1 + s_2)^2 - s_2 s_3 \\
&\geq (s_1 + s_2)(s_1 + s_2 + s_3) - (s_1 + s_2)^2 - s_2 s_3 \\
&= s_1 s_3
\end{aligned}
$$

where for the inequality, we used the fact that $k \geq s_1 + s_2 + s_3$. Finally, we now get a contradiction of the optimality of $\mathbf{S}$:
$$|S_1 \cup S_2 \cup S_3| \Pr(Y \notin S_1 \cup S_2) - |S_3| \Pr(Y \notin S_1) \geq s_1 s_3 \frac{1 - \alpha}{k - 1} > 0$$

implies
$$\langle p, \ell(\mathbf{S}) \rangle - \langle p, \ell(\mathbf{S}') \rangle > 0.$$

This proves the claim that if $\mathbf{S} = (S_1, \ldots, S_l) \in \gamma(p)$, then $l = 2$ and so $\mathbf{S} = (S_1, [k] \setminus S_1)$. Next, we show that $S_1 = \{1\}$. We already have shown that $1 \in S_1$. We calculate
$$
\begin{aligned}
\langle p, \ell((S_1, [k] \setminus S_1)) \rangle &= |S_1| - 1 + k \Pr(Y \notin S_1) \\
&= |S_1| - 1 + k(k - |S_1|)\left(\frac{1 - \alpha}{k - 1}\right) \\
&= |S_1|\left(1 - k\left(\frac{1 - \alpha}{k - 1}\right)\right) + C
\end{aligned}
$$

where $C = -1 + k^2 \left(\frac{1-\alpha}{k-1}\right)$ does not depend on $|S_1|$. To prove that $|S_1| = 1$, by minimality of $\mathbf{S}$ it suffices to show that
$$1 - k\left(\frac{1 - \alpha}{k - 1}\right) > 0.$$

To see this, note that
$$
\begin{aligned}
1 > k\left(\frac{1 - \alpha}{k - 1}\right) &\iff \frac{1}{k} > \frac{1 - \alpha}{k - 1} \\
&\iff \frac{k - 1}{k} = 1 - \frac{1}{k} > 1 - \alpha \\
&\iff \alpha > \frac{1}{k}
\end{aligned}
$$

where the last line is part of our assumption in the lemma statement. $\qquad\square$

## S6 Derivation of the figures

We discuss how Figures 1 and 2 are obtained.

### S6.1 Figure 1 from the main article

When $k = 3$, there are 12 nontrivial ordered partitions. Below, we represent $\mathcal{OP}_3$ vectorially in $\mathbb{R}^3$ using Proposition 3.6:

```
OPk = [-2  -2  -1  -1  -1  -1   0  -1   0   0   0   0 ;
        0  -1   0   0   0  -1   0  -2  -1  -1  -1  -2 ;
       -1   0   0  -1  -2   0  -1   0   0  -1  -2  -1 ]
```

Every column of the matrix `OPk` is a nontrivial ordered partition, e.g., the first column $\begin{bmatrix} -2 \\ 0 \\ -1 \end{bmatrix} \mapsto 2|3|1$.

Consider the following matrix whose columns are $\ell(\mathbf{S}) = L^{WW}(\varphi(\mathbf{S})) \in \mathbb{R}^3_+$ where $\ell$ is the ordered partition loss and $\mathbf{S} \in \mathcal{OP}_3$.

```
ell = [ 5   5   4   3   2   3   1   2   1   0   0   0 ;
        0   2   1   0   0   3   1   5   4   3   2   5 ;
        2   0   1   3   5   0   4   0   1   3   5   2 ]
```

For example, the first column of `ell` is the result of applying $L^{WW} : \mathbb{R}^k \to \mathbb{R}^k_+$ to the first column of `OPk`, *i.e.*, $\begin{bmatrix} 5 \\ 0 \\ 2 \end{bmatrix} = L^{WW}\left(\begin{bmatrix} -2 \\ 0 \\ -1 \end{bmatrix}\right) = \ell^{\mathcal{OP}}(2|3|1)$. Finally, to get the region in Figure S1 labelled by "2|3|1", we plot the $(p_2, p_3)$ coordinates of the following polytope:

$$\mathrm{Reg}(2|3|1) := \{p \in \Delta^3 : \langle p, \ell(2|3|1) - \ell(\mathbf{S})\rangle \leq 0,\ \forall \mathbf{S} \in \mathcal{OP}_3,\ \mathbf{S} \neq 2|3|1\}.$$

Repeat this procedure for all of $\mathcal{OP}_3$, we obtain Figure S1.

Figure S1: Each polygonal region is the polytope $\mathrm{Reg}(\mathbf{S})$ projected onto its last two coordinates overall $\mathbf{S} \in \mathcal{OP}_3$.

### S6.2 Figure 2 from the main article

For the left panel of Figure 2, we compute $\Omega_{LWW}$

$$\Omega_{LWW} := \{p \in \Delta^k : |\arg\max p| = 1,\ \arg\max v = \arg\max p,\ \forall v \in \Gamma_{LWW}(p)\}.$$

Thus, the region in light gray in the left panel of Figure 2 is the union of the polygons of Figure 1 labelled by an ordered partition whose the top bucket has 2 elements. This characterize $\Omega_{LWW}$ up to a set of Lebesgue measure zero.

For the right panel, consider $v \in \Gamma_{LCS}(p)$. Liu [4, Lemma 4] states that if $\max p < 1/2$, then $v = (0, 0, 0)$. Furthermore, if $\max p > 1/2$, then $\arg\max v = \arg\max p$. This characterize $\Omega_{LCS}$ up to a set of Lebesgue measure zero.