[Reviews · NeurIPS 2020]

Review 1

Summary and Contributions: The paper demonstrates that the Weston-Watkins SVM, which is known to be inconsistent for the 0-1 loss, is calibrated with respect to a discrete loss measuring the error between ordered partitions and labels. They argue (using a computational search for small number of labels) that this discrete loss is maximally informative among all calibrated discrete losses. They use their analysis to give an argument to justify the experimental superiority of this SVM over the Crammer-Singer SVM.

Strengths: This has to be categorized as a theoretical paper. The paper gives a novel understanding of the Weston-Watkins SVM by showing that it is calibrated w.r.t. a discrete loss on ordered partitions. This new understanding provides the technical tools to the authors to theoretically justify why the WW-SVM performs better in practice than the CS-SVM, which was noted by Dogan et al. 2016.

Weaknesses: The significance of the result is low for the NeurIPS conference. The authors design a discrete loss whose Bayes risk matches the Bayes risk of the WW-SVM. Although this implies that both losses are calibrated, this is still far from an explicit statement. More specifically, their result shows that both losses are calibrated w.r.t. some decoding function $\psi$ (I use the notation from their Definition 1.1), that they *do not* specify. This does not allow them to derive any explicit surrogate method, and neither allows them to provide a comparison inequality between the excess risk of the discrete loss and the excess risk of the surrogate loss, as it is done for instance for the Crammer-Singer SVM and the abstain loss in (Ramaswamy et. al. 2018), which the authors cite as a reference paper.

Correctness: The claims and method seems correct.

Clarity: The paper is in general well-written. I would suggest to the authors to try to remove some definitions/lemmas/propositions and to put more text in the main body to make it more readable.

Relation to Prior Work: Comparison with previous work is correct.

Reproducibility: Yes

Additional Feedback: The paper is clear and all statements are properly justified. However, my concern is about the significance of the theoretical result, as it is far to provide a full surrogate analysis of the WW hinge loss. I have the following questions to the authors: - Is it possible (and easy) to derive an explicit and easy-to-use decoding function $\psi$ that calibrates the WW loss to the ordered partition loss? - If this is the case, how easy would be then to derive a comparison inequality between excess risks akin to the one of Ramaswamy et. al. 2018? - You justify that WW performs better than CS using a consistency argument, i.e., saying that the measure of conditional probabilities for which the method is consistent is strictly larger than the one for CS. Can you explain why WW performs better than Lee-Lin-Wahba (LLW)?


Review 2

Summary and Contributions: The paper introduces a discrete loss (ordered partition loss) and proves that the Weston-Watkins (WW) hinge loss is calibrated w.r.t. the proposed discrete loss. It also shows that, when combined with the usual argmax decision rule, the Weston-Watkins hinge loss tends to outperform the Crammer-Singer (CS) hinge loss. The contributions of this work are, therefore, purely theoretical.

Strengths: The paper provides stronger theoretical foundations for the choice of the WW hinge loss over other multiclass losses for the SVM. These observations are consistent with previous empirical results [1]. The paper is technically sound and rigorous and the mathematical derivations seem correct (although I did not review them thoroughly).

Weaknesses: In their response letter, the authors have clarified most of my questions and have agreed to include those clarifications in the manuscript. If they do make those changes, then I have no objection regarding the presentation of this work at NeurIPS.

Correctness: The derivations are quite technical and I did not review them exhaustively. Still, I found no major or obvious mistake, everything seems correct.

Clarity: The paper is very well written and I did not find any typo. As argued before, authors should try to enhance clarity by providing a more intuitive explanation of some of the presented results and definitions.

Relation to Prior Work: The paper presents a clear review of relevant prior work in the field and explains how it relates with their current work. As I am not an expert in this particular subject, I am unaware if any relevant references are missing.

Reproducibility: Yes

Additional Feedback:


Review 3

Summary and Contributions: The author presented a theoretical analysis of the Weston-Watkins (WW) hinge loss surrogate for multiclass classification. Previous research by Dogan et al. (2015) shows that even though the WW surrogate is not Fisher consistent with respect to the 0-1 loss, the experiments show its empirical benefit compared to other formulations, including the Fisher consistent ones. The author showed a new theoretical insight on why the WW surrogate performs well in practice. The author proved that the WW surrogate is a Fisher consistent surrogate over the ordered partition loss, a discrete ranking-like loss that the author defined. The technique that the author used to prove the result is based on recent results by Finocchiaro et al. (2019) on polyhedral surrogates. In addition, the author also provided new insight on why the WW surrogate performs better than the CS surrogate when the problems do not have a dominant label.

Strengths: Overall, I like the paper since it fills a gap in our understanding of multiclass SVM formulations. It has always been a mystery why the WW surrogate loss performs well empirically despite lacking nice theoretical property. The techniques that the author uses for the theoretical analysis (as well as the technique in Finocchiaro et al. (2019)) is also interesting. === Post rebuttal === Thank you for the author for providing the feedback. I have read it as well as other reviews. I keep my acceptance score unchanged. I would also suggest the author to include the discussion in Fathony et al, (2016), about the absolute and relative margin formulations of multiclass SVM in the revision. Particularly, they proposed a consistent relative margin formulation to avoid the LLW's pitfall of using the absolute margin formulation.

Weaknesses: Even though this result provides a nice theoretical justification for the WW surrogate, few questions remain. I would like to ask the author to comment on these questions. 1) The author shows that the WW surrogate is Fisher consistent over the ordered partition loss. However, since the experiments show that The WW surrogate performs well on the 0-1 loss, a question arises on how the performance over the ordered partition loss translates to the 0-1 loss. In particular, since the orderer partition loss that the author proposed is very different from the 0-1 loss. I suggest the author to explain the connection in the revised paper. 2) The author suggested that one of the reasons for the WW success (compares with the CS surrogate) is that the \Omega of the WW surrogate is a strict superset of the \Omega of the CS. These additional areas include cases where there is no majority label. Even though it (partially) explain the success of the WW surrogate in the case where no majority label, it does not explain why the LLW surrogate (the Fisher consistent one) does not perform as well as the WW in some cases. The \Omega of the LLW surrogate is a strict superset of the \Omega of the WW, yet it does not translate to better performances. Could the author provide a comment on the success of the WW surrogate despite its \Omega is a strict subset the LLW one, or from the other perspective, why the LLW surrogate fails even though its \Omega is a strict superset of the WW one. 3) Even though it is not required for a purely theoretical paper, having a small empirical result will be nice. In particular, I am interested to see the empirical performance of different surrogates on the discrete ordered partition loss that the author proposed. This may provide an empirical insight on the superiority of the WW on the loss metric as well as how the performance on the metric translates to the performance on the 0-1 loss metric.

Correctness: The methodology proposed in the paper looks correct. However, I did not thoroughly check the proofs of the theorems.

Clarity: Overall, the paper is well written with clear notations and formulations.

Relation to Prior Work: Yes. All the related contributions that I am aware of are discussed in the paper.

Reproducibility: Yes

Additional Feedback:


Review 4

Summary and Contributions: The paper analyzes surrogate loss optimized by Weston-Watkins formulation of multi-class SVMs. It is shown that the Weston-Watkins (WW)-loss is calibrated with so called ordered partial loss which is a discrete loss introduced by the authors. The authors further show conditions on the posterior distribution under which the minimizer of WW-loss processed by "argmax" operator provides Bayes optimal decision under 0/1-loss. It is shown that this set of distributions subsumes those for which Crammer-Singer surrogate is optimal.

Strengths: The fact that WW-loss is not calibrated with 0/1-loss has been in contradiction to empirical findings that WW multi-class SVMs perform well in practice. The paper provides theoretically grounded results that help to resolve this discrepancy to some extent.

Weaknesses: The introduced ordered partial loss, the core concept of the paper, is hard to grasp intuitively. It is not 100% clear why this discrete loss should be the target objective to optimize.

Correctness: Yes.

Clarity: Very clearly written.

Relation to Prior Work: Provided relation to existing works is exemplary.

Reproducibility: Yes

Additional Feedback: A natural question is how to get the top bucket (or ideally the whole sequence of buckets) from the surrogate decision function (Theorem 5.2. answers this question partially). If it is not possible for whatever reason this should be explicitly stated. I read the authors' feedback.

[Author Response · NeurIPS 2020]

We thank the reviewers for the thought-invoking questions and helpful comments on improving the manuscript. We will
add more text to improve readability and, if necessary, remove some of the technical lemmas to make room. We also
have extended our empirical proof of maximal informativeness to $k = 15$.

**R1, R2, & R3:** The LLW hinge loss is calibrated with respect to the 0-1 loss while the WW hinge loss is not. Why
does the WW SVM still outperform the LLW SVM? In other words, how can calibration be used as a justification for
performance?

**A:** The LLW SVM performs worse for a reason unrelated to calibration. Doğan et al. [2016] makes the distinction
between *relative* and *absolute* margin losses. The WW and CS loss are both based on relative margins, while LLW is
based on absolute margins. Doğan et al. [2016] on their page 20 gave an explanation for the worse performance of all
losses based on absolute margin. Hence, the poor performance of LLW is a consequence of using absolute margin. Out
of the nine SVM losses considered by Doğan et al. [2016], only the CS and WW losses are relative margin based. We
will add this discussion to our manuscript.

**R2, R3 & R4:** Why is consistency with respect to the ordered partition loss desirable? What is the intuition?

**A:** Regarding the intuition behind the ordered partition loss, the basic idea is that we want to rank the labels,
where ties are allowed and each $S_i$ is a set of labels that are tied. We want the correct label to be as high
up the ranking as possible. The lower the true class is ranked, the larger the loss. That is what the definition of
the ordered partition loss says. We should have said this in the initial draft and will add this discussion to our manuscript.

**R1 & R4:** How to get the surrogate decision function $\psi$ to recover the ordered partition/buckets? Is there an excess
risk bound?

**A:** In line 125 of our manuscript, we cited Finocchiaro et al. [2019] who provided an explicit $\psi$ given $L$, $\ell$, and $\varphi$. We
refer to [Finocchiaro et al., 2019, Definition 6] for the construction of $\psi$. The excess risk bound for their constructed
$\psi$ can be found in [Finocchiaro et al., 2019, Theorem 6]. We will make the theorem references explicit in the manuscript.

**R2:** What are the consequences of the maximally informative property?

**A:** Intuitively, a discrete loss $\ell : \mathcal{R} \to \mathbb{R}_+^k$ (where $\mathcal{R}$ is finite) with embedding $\varphi$ (an injection into the domain
of $L$) is maximally informative for a surrogate $L$ if $\varphi(\mathcal{R})$ captures all the essential information contained in the
surrogate $L$ in the most compact way. To better convey this intuition, we replace "maximally informative" with the new
terminology "minimally emblematic." Let us say that a set of vectors $E \subseteq \mathbb{R}^k$ is an *emblem* of $L$ if for all $p \in \Delta^k$, the
set $E \cap \operatorname{argmin}_v \langle p, L(v) \rangle$ is nonempty. Then we can equivalently define $\ell$ with $\varphi$ to be *minimally emblematic* for $L$ if
$\varphi(\mathcal{R})$ is an emblem of $L$ of minimal cardinality. In other words, $\varphi(\mathcal{R})$ is a minimal set of minimizers of all possible
$L$-inner risks. We will update our manuscript with this discussion and the new terminology.

**R3:** How does performance over the ordered partition loss translate to the 0-1 loss?

**A:** Results from our section on the "argmax link" provide a partial answer to this. Namely, we show in two common
regimes, the Bayes optimal ordered partition has a top bucket consisting of a single element. When this occurs, the
argmax link recovers the most probable class, i.e., the unique element from the top bucket. We will modify the
manuscript to clarify this point.

# 41 References

U. Doğan, T. Glasmachers, and C. Igel. A unified view on multi-class support vector classification. *The Journal of
Machine Learning Research*, 17(1):1550–1831, 2016.

J. Finocchiaro, R. Frongillo, and B. Waggoner. An embedding framework for consistent polyhedral surrogates. In
*Advances in neural information processing systems*, pages 10780–10790, 2019.


[Meta-Review · NeurIPS 2020]

This is a nice theoretical contribution to the field of multiclass classifiers, proving that a well known loss (the so called Weston-Watkins loss, defined as the sum of hinge losses over relative log-odd scores) is calibrated for a new discrete loss introduced in the paper. While the practical implications of the results are not direct (and in particular, how this result explains the good empirical performance of the WW loss), it provides new perspectives and solid foundations to analyze loss functions for multiclass problems. The paper is overall very clear, rigorous and polished.